# The tumor-sentinel lymph node immuno-migratome reveals CCR7+ dendritic cells drive response to sequenced immunoradiotherapy

Surgical ablation or broad radiation of tumor-draining lymph nodes can eliminate the primary tumor response to immunotherapy, highlighting the crucial role of these nodes in mediating the primary tumor response. Here, we show that immunoradiotherapy efficacy is dependent on treatment sequence and migration of modulated dendritic cells from tumor to sentinel lymph nodes. Using a tamoxifen-inducible reporter paired with CITE-sequencing in a murine model of oral cancer, we comprehensively characterize tumor immune cellular migration through lymphatic channels to sentinel lymph nodes at single-cell resolution, revealing a unique immunologic niche defined by distinct cellular phenotypic and transcriptional profiles. Through a structured approach of sequential immunomodulatory radiotherapy and checkpoint inhibition, we show that sequenced, lymphatic-sparing, tumor-directed radiotherapy followed by PD-1 inhibition achieves complete and durable tumor responses. Mechanistically, this treatment approach enhances migration of activated CCR7+ dendritic cell surveillance across the tumor-sentinel lymph node axis, revealing a shift from their canonical role in promoting tolerance to driving antitumor immunity. Overall, this work supports rationally sequencing immune-sensitizing, lymphatic-preserving, tumor-directed radiotherapy followed by immune checkpoint inhibition to optimize tumor response to immunoradiotherapy by driving activated dendritic cells to draining sentinel lymph nodes.

Head and neck squamous cell carcinoma (HNSCC), often caused by tobacco, ethanol, other carcinogens, as well as human papillomavirus, and other oncogenic viruses, typically presents in an advanced stage (III–IV) and claims the lives of nearly half of all affected patients[1,2]. Primary surgery and/or radiotherapy with cytotoxic chemotherapy have long been the mainstays of curative-intent therapies for locally advanced HNSCC and yield only modest improvements in cure rates over the past few decades[3]. Immunotherapy with PD-1 inhibitors has emerged as a new standard of care for recurrent/metastatic HNSCC,

with significant improvement in overall survival and dramatic long-term control for a minority of patients. However, recent clinical trials adding concurrent and adjuvant PD-1 inhibition to standard of care chemoradiation treatment of primary tumor and draining lymphatic basins in a previously untreated, locally advanced setting have yielded disappointing results with no oncologic benefit[4–6]. These findings suggest that the current standard therapy directed at draining lymphatics, sequenced concurrently or prior to immunotherapy in the locally advanced setting, may compromise the host antitumor immune

✉ e-mail: rsaddawi@health.ucsd.edu; jcalifano@health.ucsd.edu

response, inhibiting response to checkpoint inhibition. Recent phase I/II trials with a variety of designs have shown that sparing uninvolved lymphatics facilitates enhanced—and in some cases dramatic—responses to tumor-directed radiotherapy in combination with checkpoint inhibition[7–9]. These findings suggest that preserving lymphatic structures may play a critical role in coordinating effective antitumor immunity. Yet, the mechanisms by which lymphatic sparing enhances response remain poorly defined. In particular, the role of immune cell migration from tumors to lymphatics in driving these enhanced responses remains inadequately understood.

Our previous work demonstrated the critical role of tumor-draining lymph nodes in mediating the host response to immunotherapy, indicating that effective antitumor immunity requires intact and functional regional lymphatics[10]. Here, we hypothesize that appropriate sequencing and timing of tumor-directed immunomodulatory radiotherapy and immunotherapy can enhance tumor response. Our findings reveal that lymphatic-sparing, tumor-directed radiotherapy followed by PD-1 inhibition achieves complete and durable tumor responses by enhancing dendritic cell migration across the tumor-sentinel lymph node axis, suggesting that lymphatic-sparing immunomodulatory therapies may convert migratory dendritic cells from a pro-tolerogenic to antitumor phenotype. Mechanistically, we demonstrate the requisite role of CCR7+ migratory dendritic cells, which necessarily transit through afferent lymphatics to sentinel nodes, for both the clonotypic expansion of antitumor T cells and the successful tumor response to immunoradiotherapy. These findings highlight the therapeutic potential of preserving functional draining lymph nodes to enhance immune cell migration and improve antitumor responses.

## Results

### Sentinel lymph node mapping in tumor-bearing mice

Clinical studies have highlighted the importance of the sentinel lymph node (SLN) in various cancers, including melanoma, breast cancer, and HNSCC. Sentinel node biopsy has proven to be a valuable tool for staging and treatment planning in these cancers, offering less invasive alternatives to traditional neck dissection while maintaining high predictive accuracy for metastasis[11–15]. More recently, there is evidence suggesting that the SLN may serve as an essential hub for initiating and coordinating antitumor immune responses and, thereby, play a pivotal role in cancer immunology[16,17].

Given the sentinel lymph node's pivotal role in cancer immunology, we extended our previous work on tumor-draining lymph nodes to specifically focus on sentinel lymph nodes[10]. To understand the dynamics of immune cell trafficking in the context of antitumor immunity, we mapped the sentinel lymph nodes in our orthotopic, syngeneic 4MOSC murine models—both the checkpoint inhibitor sensitive (4MOSC1) and insensitive (4MOSC2) models—which mimic tobacco-associated HNSCC and afford a valuable platform for studying the dynamics of immune cell trafficking[18]. Utilizing Isosulfan blue, a dye that binds to interstitial serum albumin and is carried predominantly by lymphatics, and Tilmanocept IRDye-800CW as lymphatic tracers, we accurately identified the draining SLNs relative to the tumor and other anatomical landmarks within 30 minutes of primary tumor injection, (Supplementary Fig. 1A–F). This dual-tracer technique enabled precise mapping and confirmation of SLN locations using both visual and fluorescent imaging modalities (Supplementary Fig. 1A, B). Our quantification revealed that 59% of the tumor-bearing mice exhibited ipsilateral SLN localization; however, 34% and 13% of mice showed contralateral and bilateral SLN localization, respectively (Supplementary Fig. 1D). These data indicate that simply harvesting unmapped ipsilateral lymph nodes is unlikely to identify key sites of lymphatic drainage and the primary SLN drainage niche for migratory immune cells. This approach ensured precise identification and characterization of the SLN immunologic niche, and demonstrated

concordance with clinical findings in HNSCC, in which SLN have variable drainage patterns in individual patients, and accurate SLN mapping is paramount for staging and treatment planning[11,13,15]. This provided a robust framework for studying locoregional immune dynamics and migration of immune cells from the primary tumor to the first echelon draining sentinel nodes. Importantly, in subsequent experiments, we exclusively used isosulfan blue to avoid any inadvertent in vivo influence from Tilmanocept, which binds to mannose-binding proteins often displayed on myeloid-lineage immune effectors.

### Defining the tumor-sentinel lymph node immune migratome

To further characterize the cellular composition within tumor-draining SLNs, we utilized an inducible Cre reporter model system, allowing for precise temporal control of tdTomato fluorescent protein expression upon tamoxifen induction. Specifically, we employed the R26-CreERT2 x Ai9 mouse model, where Cre expression is under the control of the ubiquitous Rosa promoter, ensuring that all cells are responsive to tamoxifen and capable of expressing the tdTomato marker (Fig. 1A). Additionally, we used the e8i-CreERT2 x Ai9 reporter animals, in which tamoxifen-induced tdTomato expression is restricted to CD8 T cells, allowing us to specifically track the migration of these cells (Supplementary Fig. 1F). While local tamoxifen dosing in the oral cavity led to tdTomato+ cell detection within the SLN at 48 hours, removing the inoculation site (partial glossectomy) 2 hours after tamoxifen injection resulted in the absence of tomato red cells in SLN, indicating that tamoxifen diffusion through lymphatics did not contribute to tdTomato expression (Fig. 1B). To determine whether tamoxifen might drain to the SLN and label cells outside of the tumor over an extended period of time, and to test this in the context of an active host response to a local immune stimulus, we employed a model of localized immune activation associated with immune effector trafficking from the periphery to draining lymph nodes[19,20], in which oral inoculation with the surrogate antigen ovalbumin along with adjuvant induces a localized, antigen-specific inflammation in vivo[10]. Using this model in e8i-CreERT2 x Ai9 reporter animals, we found that surgical interruption of the afferent lymphatic channel to the draining SLN 72 hours after inoculation was sufficient to block trafficking of tdTomato+ CD8 + T cells, confirming a unidirectional and requisite migration from the peripheral site of inflammation to the SLN (Fig. 1C). Importantly, these findings demonstrate that tdTomato labeling in the SLN does not result from delayed passive diffusion of tamoxifen in the setting of localized oral cavity inflammation.

R26-CreERT2 x Ai9 oral tumor-bearing mice treated locally with tamoxifen demonstrated increased tdTomato+ cells in the SLN compared non-SLN controls and spleen 72 hours after labeling (Fig. 1D). To assess whether migratory immune cells from the tumor exhibit not only greater abundance but also a distinct composition within the SLN, we compared tdTomato+ CD45+ populations across nodal compartments. This analysis revealed a unique immune profile in the SLN, compositionally distinct from that observed in non-sentinel lymph nodes (Fig. 1E). Quantitative analysis revealed selective enrichment of B cells, dendritic cells, MHCII[hi] activated dendritic cells, and progenitor-exhausted (Tpex) CD8 + T cells in the SLN (Fig. 1F; gating strategy as in Supplementary Fig. 2A), whereas naive-like CD8 + T cells were more prevalent in non-SLNs (Supplementary Fig. 2B). These findings support the interpretation that the SLN represents a distinct immunologic site within the tumor-draining axis, where migratory immune cells preferentially accumulate and begin to acquire features associated with functional activation. As a final confirmation of this model, immunofluorescence of whole-mount en bloc resected tumor-lymphatic channel-sentinel lymph node preparations harvested 48 hours after local tumor tamoxifen injection in R26-CreERT2 x Ai9 mice allowed us to visually track tdTomato+ cells migrating from the

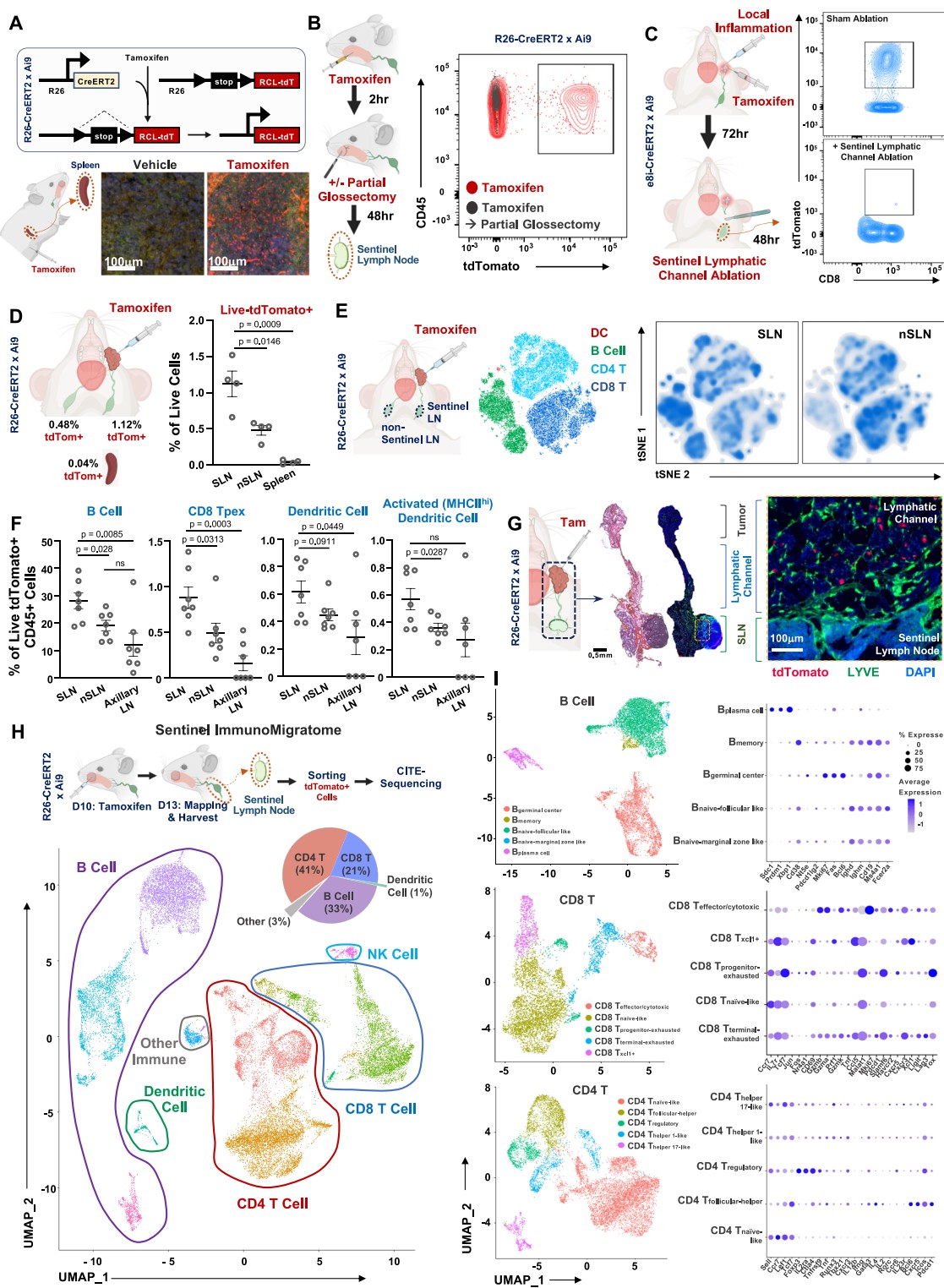

primary tumor through draining lymphatic channels directly to SLNs (Fig. 1G).

To define the identity and transcriptome of the migrating cell populations, or "migratome," from the tumor to SLNs, we performed CITE-sequencing on tdTomato+ cells isolated from the SLNs of tumor-bearing mice 72 hours after tamoxifen injection. UMAP clustering revealed an unexpectedly diverse SLN immune migratome, including distinct populations of CD4 + T cells, CD8 + T cells, B cells, dendritic cells, and NK cells (Fig. 1H, I, Supplementary Fig. 2C). These data suggest that the SLN is the main destination for a dynamic tumor immunomigratome, which is different from the immune populations within the tumor microenvironment (TME, as we previously characterized[18]). Given the diversity of this cell population and the preferential migration of immune cells to the SLN compared to other draining lymph nodes, we hypothesized that the tumor-draining SLN harbors a unique immunologic niche and may offer insights into the mechanism of host antitumor response. We then sought to examine SLN biology in the context of lymphatic-sparing, tumor-directed therapy to gain further insights into SLN contributions to tumor response.

**Fig. 1 | Host antitumor surveillance is defined by a diverse immunomigratome to the sentinel lymph node. A** Cartoon schema for genetically engineered reporter animal model: the R26-CreERT2 x Ai9 reporter mice, which express tdTomato fluorescent protein upon tamoxifen induction. The schematic illustrates the genetic strategy for Cre-mediated tdTomato expression in these mice. Mice were dosed with tamoxifen at 100 mg per kg (body weight) intraperitoneally. Representative immunofluorescence images show increased tdTomato+ cells in the spleen from tamoxifen-treated spleen 72 hours after systemic delivery of tamoxifen. Representative of $n = 3$ biologically independent samples; experiment was independently repeated at least twice with similar results. **B** (left) Cartoon schema illustrating the experimental model in which tamoxifen was injected intraorally 0.05 mg in 2.5 μL for intratumoral injection, followed by partial glossectomy after 2 hours and then delayed sentinel lymph node harvest after 48 hours. (right) Flow cytometry analysis of sentinel lymph nodes from locally dosed tamoxifen in the tongues of R26-CreERT2 x Ai9 reporter animals that were treated with either sham surgery or partial glossectomy, showing tdTomato+ CD45+ cells. This demonstrates a lack of tdTomato+ cells in SLN at 48 h after delayed partial glossectomy 2 hours after injection, indicating a lack of tamoxifen diffusion into SLN. Representative of $n = 3$ biologically independent samples; experiment was independently repeated at least twice with similar results. **C** (left) Cartoon schema for experimental model in which the sentinel lymphatic channel was interrupted in e8i-CreERT2 x Ai9 reporter animals in which local inflammation was modeled in the oral cavity with injection of Ovalbumin and STING agonist, delivered concurrently with local tamoxifen. Animals then underwent either sham surgery or surgical ablation of the sentinel lymphatic channel 72 hours later, and SLNs were harvested 48 hours after surgery to assess tdTomato+ CD8 + T cell trafficking. (right) Flow cytometry of sentinel lymph nodes from locally dosed tamoxifen in the buccal space of e8i-CreERT2 x Ai9 reporter animals that were treated with either sham surgery or sentinel lymphatic channel ablation in the context of local oral inflammation. Interruption of the afferent lymphatic channel to the SLN was sufficient to block the trafficking of CD8 T cells in e8i-CreERT2 x Ai9 reporter animals. Representative of $n = 4$ biologically independent samples; the experiment was independently repeated at least twice with similar results. **D** Quantification of live tdTomato+ cells in SLNs, non-SLNs, and spleen of tamoxifen-treated mice, showing the highest presence in SLNs. Tumor-bearing mice were dosed with 0.05 mg tamoxifen in 2.5 μL miglyol intratumorally followed by tissue harvest for flow cytometry 72 hours after delivery of tamoxifen. Data are presented as mean values ± SEM ($n = 4$ biologically independent samples/group); $p$ values calculated by two-sided unpaired Student's $t$ test. **E** (left) Cartoon schema illustrating the experimental model in which tamoxifen was injected intraorally

into 4MOSC1 tumor-bearing R26-CreERT2 x Ai9 reporter mice, followed by sentinel lymph node harvest at 72 hours. (right) tSNE clustering of tdTomato+ CD45+ immune cells from SLNs versus nSLNs shows a compositionally distinct population in the SLN; tSNE, t-distributed stochastic neighbor embedding, was used for dimensionality reduction and visualization of immune cell populations (see methods). Representative of $n = 7$ biologically independent samples; the experiment was independently repeated at least twice with similar results. **F** Quantification of tdTomato+ immune cell subsets from SLNs, nSLNs, and axillary lymph nodes in 4MOSC1 tumor-bearing R26-CreERT2 x Ai9 reporter mice, harvested 72 hours after intratumoral tamoxifen injection. Selective enrichment of B cells, dendritic cells, MHCII$^{hi}$ activated dendritic cells, and progenitor-exhausted CD8 + T cells is observed in SLNs. Data are presented as mean values ± SEM ($n = 7$ biologically independent samples/group); $p$ values calculated by two-sided unpaired Student's $t$ test. **G** Following intratumoral injection of with 0.05 mg tamoxifen in 2.5 μL miglyol, tdTomato+ cells were tracked from the tumor site to the SLNs. Tissues were harvested for analysis at 48 hours following local delivery of tamoxifen. (Top) Cartoon schema and representative H&E and immunofluorescence images of en bloc resected tumor, afferent lymphatic, and sentinel lymph node specimen; (bottom) Representative immunofluorescence imaging of the lymphatic channel and SLNs with tdTomato+ cells tracking along lymphatic vessels adjacent to the SLN. Representative of $n = 4$ biologically independent samples; the experiment was independently repeated at least twice with similar results. **H** The Cancer Immunomigratome. 4MOSC1-tumor bearing animals were injected intratumorally with 0.05 mg tamoxifen in 2.5 μL Miglyol 10 days after tumor transplantation, followed by sentinel lymph node mapping and harvest 72 hours after delivery of tamoxifen. Subsequently, CITE-sequencing was performed on tdTomato+ cells isolated from SLNs to characterize immune cell populations involved in antitumor surveillance. UMAP clustering revealed a diverse immunomigratome, including distinct populations of CD4 + T cells, CD8 + T cells, B cells, dendritic cells, NK cells, and other immune cells. Nested pie chart shows the relative abundance of each major migratory population as a percent of the total. $n = 2$ biologically independent samples/group. **I** UMAP subclustering and dot plots of tdTomato+ B cells, CD8 + T cells, and CD4 + T cells from SLNs 72 hours after intratumoral tamoxifen injection from **H** above. Expression patterns of canonical surface markers support classification of migratory immune subsets as defined in the figure. $n = 2$ biologically independent samples/group-Created in BioRender. Saddawi-Konefka, R. (2025) https://BioRender.com/emood6a, https://BioRender.com/9yg4is9, https://BioRender.com/m675tdd. Source data are provided as a Source Data file.

## Draining lymph node sparing, tumor-directed radiation is immunomodulatory

Radiation therapy has been shown to influence immune responses in the TME, enhancing the efficacy of immunotherapy through various mechanisms, including the modulation of immune cell trafficking and activation[21,22]. Previous studies have demonstrated that the immunomodulatory effects of radiotherapy are dose-dependent, with radiation promoting immune cell activation and migration while minimizing cytotoxicity[23–26]. Conversely, high-dose radiation, with bystander damage to peritumoral healthy tissues, can lead to lymphodepletion and increased immunosuppressive cell populations, such as regulatory T cells and tumor-associated macrophages, which can inhibit the antitumor immune response[27–29].

We designed a lymphatic-sparing, immunomodulatory radiation therapy strategy to investigate the biology of immune cell migration and immunosurveillance to the SLN in response to immunoradiotherapy treatment. To examine the effects of tumor-directed radiation therapy (tdRT) on local immunosurveillance, we evaluated the response of 4MOSC1-tumor-bearing animals to tdRT that spared draining lymphatic basins (Fig. 2A, Supplementary Fig. 3A, B). A single fraction of 4 Gy of tumor-targeted radiation was found to be subcytotoxic in both 4MOSC1 and 4MOSC2 models, but highly immunomodulatory (Fig. 2B–D, Supplementary Fig. 3C). Notably, pathway enrichment analysis of immune-related gene expression in tumors following tdRT identified significant upregulation of key pathways

involved in chemokine response, myeloid leukocyte migration, and regulation of inflammatory response (Fig. 2E). These findings suggest potent immunomodulatory effects of tdRT, particularly in pathways associated with antitumor immunosurveillance, and support early activation programs in migratory dendritic cells.

To further investigate the immunologic consequences of fractionated tdRT, we compared single (4 Gy × 1) and double (4 Gy × 2) fraction regimens. Although 4 Gy × 2 induced tumor rejection, the frequency of CCR7+ DCs trafficking to SLNs remained comparable to that observed with a single 4 Gy dose (Supplementary Fig. 3D, E). This suggests that the immune-modulatory effects of tdRT on DC migration may saturate after a single exposure, with subsequent fractions potentially exerting effect through other cytotoxic mechanisms.

Further analysis corroborated tdRT-induced immune cell trafficking within the TME. Flow cytometric analysis demonstrated increased trafficking of CXCR3 + CD8 + T cells and MHCII + CCR7+ dendritic cells (DCs) post-tdRT (Fig. 2F, gating strategy in Supplementary Fig. 2B and 4D), in addition to other immune effector lymphocyte populations (Supplementary Fig. 4A–C). Concomitantly, non-immune cells within the TME exhibited reduced CD47 expression following tdRT, potentially reducing barriers to immune cell phagocytosis (Fig. 2G). Furthermore, an increase in the cross-presentation of the model antigen ovalbumin on antigen-presenting cells was observed using our 4MOSC1-LucOS model, which expresses

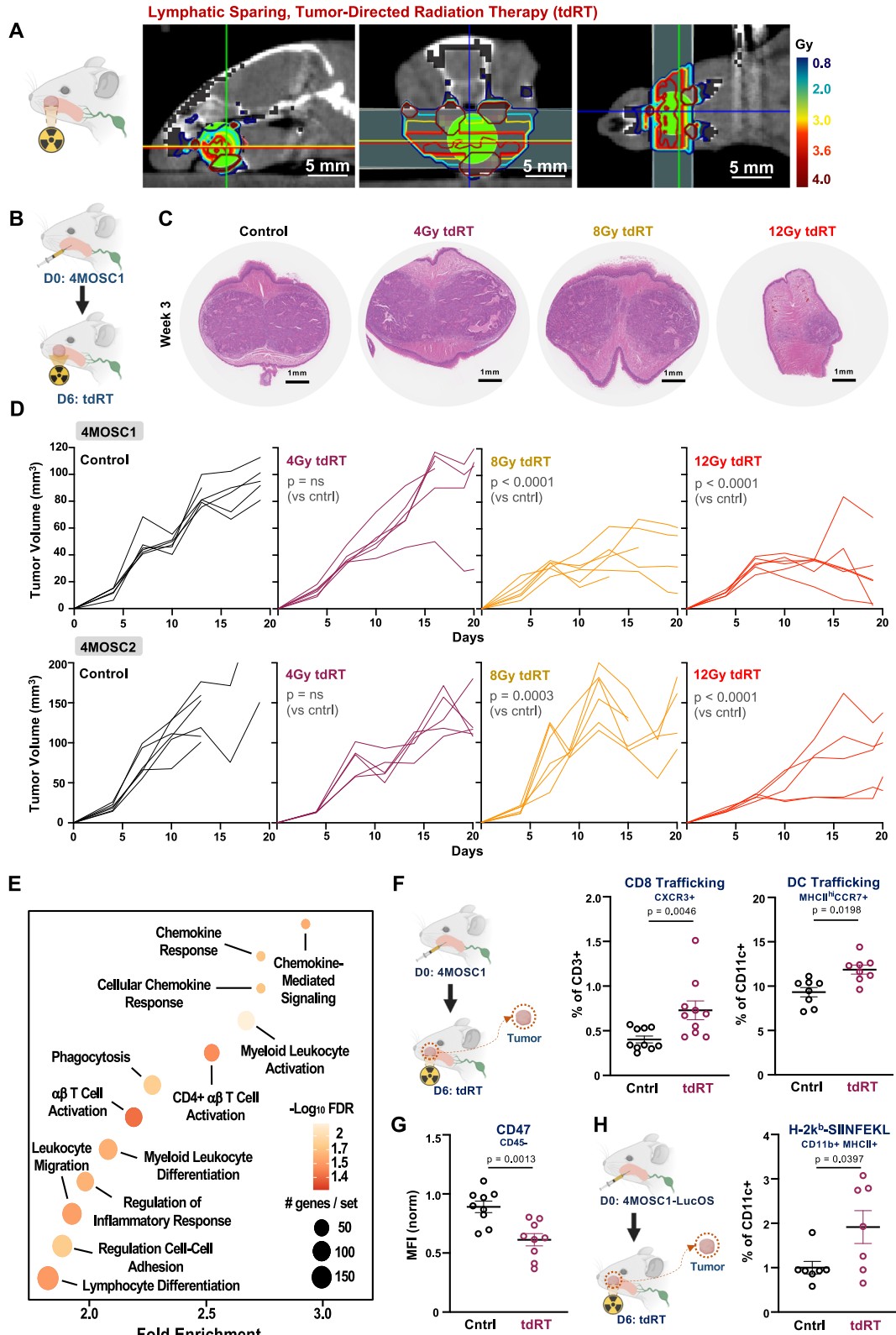

luciferase and ovalbumin, serving as a model antigen to track immune responses in vivo (Fig. 2H; gating strategy in Supplementary Fig. 4D)[10,30]. These results collectively suggest that tumor-directed, lymphatic-sparing radiation enhances local immunosurveillance within the tumor microenvironment by promoting immune cell trafficking and antigen presentation, thereby potentiating anti-tumor immune responses.

## Tumor-directed radiation primes immunologic programs for durable response to PD-1 inhibition

Finding that sub-cytotoxic tumor-directed radiation therapy (tdRT) promotes local immunosurveillance, we explored whether tdRT could potentiate immune checkpoint inhibitors (ICIs) such as αPD-1. This approach aligns with clinical trials combining targeted radiation and immunotherapy, which have shown promising results in enhancing

**Fig. 2 | Tumor-directed, non-cytotoxic radiation promotes local immuno-surveillance in the tumor microenvironment. A** (left) Cartoon schema of tumor-directed radiation therapy. (right) Representative CT images of sagittal, coronal, and axial series overlaid with radiation planning. Animals were anesthetized with isoflurane and positioned within the small animal radiotherapy machine. A spiral CT scan with 1 mm cuts of the neck was obtained, and cervical lymphatics were delineated as the planning target volume. A 5 mm collimator was installed, and two static parallel opposed beams were used to deliver homogenous single fraction doses to the planned target volume. Representative of $n = 10$ biologically independent samples; the experiment was independently repeated at least twice with similar results. **B** Cartoon schema of the experimental approach. WT animals were injected with 4MOSC1 and then treated with tdRT on day 6. **C** Representative H&E staining of tumor sections from control and tdRT-treated groups (4 Gy, 8 Gy, 12 Gy tdRT) at week 3 post-treatment. Tumor samples were fixed in zinc formalin fixative and sent for embedding, sectioning, and H&E staining. Slides were analyzed using QuPath software. Representative of $n = 3$ biologically independent samples; experiment was independently repeated at least twice with similar results. **D** Tumor growth curves for control and tdRT-treated groups (4 Gy, 8 Gy, 12 Gy tdRT) from 4MOSC1- and 4MOSC2-tumor bearing animals (***top*** and ***bottom***, respectively). Tumor volumes were measured over time, with significant differences observed at specific time points ($n = 6$ mice per group for Control, 6 mice for 4 Gy, 6 mice for 8 Gy, and 6 mice for 12 Gy, $p$ = ns for 4 Gy, $p < 0.0001$ for 8 Gy and 12 Gy). $p$ values calculated by two-sided unpaired Student's t-test. Best-fit lines and $p$ values calculated by simple linear regression (two-sided). **E** Pathway enrichment analysis of immune-related gene expression changes in tumors following tdRT. Key pathways include chemokine response, myeloid leukocyte migration, and regulation of inflammatory response. Tumors were harvested, and RNA was isolated using Qiagen RNeasy Mini Columns. Library preparation and paired-end RNA sequencing

were performed by Novogene. Gene set enrichment analysis was conducted using the GSEAPreranked module on the GenePattern public server, with the Gene Ontology (Biological Processes) and ImmunesigDB gene set collections used. X-axis represents gene sets ranked by normalized enrichment score (NES); Y-axis represents the $-\log_{10}$(FDR q-value). **F** (left) Cartoon schema of the experimental approach. WT animals injected with 4MOSC1 and then treated with tdRT on day 6 followed by tumor harvest on day 10. (right) Flow cytometric analysis of trafficking CXCR3 + CD8 + T cells and migratory MHCII + CCR7+ dendritic cell (DC) populations in the TME post-tdRT. Quantification is shown ($n = 8$–10 biologically independent samples per group, $p$ = 0.0046 for CD8 + T cells, $p$ = 0.0198 for DCs). Tumors were isolated, minced, and processed into single-cell suspensions using the Tumor Dissociation Kit and gentleMACS Octo Dissociator. Flow cytometry was performed using fluorochrome-conjugated antibodies. Data are presented as mean values ± SEM; $p$ values calculated by two-sided unpaired Student's t-test. **G** CD47 expression on tumor cells post-tdRT, measured by flow cytometry and shown as normalized median fluorescence intensity (MFI) on live CD45- cells ($n = 9$ biologically independent samples per group, $p$ = 0.0013). Data are presented as mean values ± SEM; $p$ values calculated by two-sided unpaired Student's t test. **H** (left) Cartoon schema of the experimental approach. WT animals injected with 4MOSC1-LucOS (ovalbumin expressing) and then treated with tdRT on day 6 followed by tumor harvest on day 10. (right) Flow cytometric analysis of H-2Kb-SIINFEKL expressing antigen-presenting cells post-tdRT. Quantification of normalized populations is shown ($n = 7$ biologically independent samples per group, $p$ = 0.0397). Tumors were processed as described above, and flow cytometry was performed using H-2Kb-SIINFEKL specific antibodies. Data are presented as mean values ± SEM; $p$ values calculated by two-sided unpaired Student's t-test. Created in BioRender. Saddawi-Konefka, R. (2025) https://BioRender.com/m675tdd. Source data are provided as a Source Data file.

---

antitumor responses[7–9,21,22,31,32]. To evaluate the impact of tdRT on the tumor immune microenvironment, we first examined PD-L1 expression in 4MOSC1-tumor-bearing animals treated with tdRT. Representative H&E and PD-L1 staining of tumor sections post-tdRT treatment indicated a high Combined Positive Score (CPS) for PD-L1 staining, supporting the rational addition of αPD-1 ICI in our model (Fig. 3A, B, Supplementary Fig. 5A)[33].

To examine how tdRT may prime the host for effective immune checkpoint blockade, we compared transcriptional programs across tumors treated with αPD-1 alone, tdRT alone, or tdRT followed by αPD-1. As previously shown in Fig. 2E, tdRT alone induced selective enrichment of immunologic programs, including chemokine response and inflammatory signaling, but did not broadly engage pathways associated with adaptive immunity. In contrast, αPD-1 monotherapy did not significantly alter immune-related transcriptional programs (Supplementary Fig. 5B). However, tumors treated with tdRT followed by αPD-1 exhibited marked upregulation of antigen processing and presentation, phagocytosis, and T cell activation pathways (Fig. 3C, Supplementary Fig. 5C). Direct comparison of tdRT versus combination-treated tumors further confirmed that combination therapy engaged additional immune surveillance programs beyond those induced by tdRT alone (Supplementary Fig. 5D). These data suggest that tdRT primes the TME and sensitizes tumors to immune checkpoint inhibition, further supporting a mechanistic basis for synergy when αPD-1 is administered following radiation.

Next, we assessed the therapeutic efficacy of combining tdRT with αPD-1 treatment, administering αPD-1 following tdRT to evaluate whether this sequencing improves response. WT animals bearing 4MOSC1 tumors were treated with tdRT on day 6, followed by αPD-1 administration on days 8 or 10 (Fig. 3D). Analysis of the TIME post-treatment in these groups showed increased levels of both IFNγ and IFNβ in tdRT → αPD-1 treated animals, indicative of a robust antitumor immune response (Fig. 3E). Tumor growth curves in both 4MOSC1 and 4MOSC2-tumor bearing animals demonstrated that while αPD-1 alone resulted in a modest response, the combination of tdRT and αPD-1 led to a significantly higher response rate, even in checkpoint unresponsive 4MOSC2 models (Fig. 3F). Next, we investigated the long-term

impact of combined tdRT → αPD-1 therapy in a rechallenge model in which fresh tumor was implanted in naive mice vs complete responders to tdRT → αPD-1. Tumor volume measurements revealed a significant suppression of tumor growth in complete responders compared to naive controls (Supplementary Fig. 5E). These results collectively demonstrate that sub-cytotoxic, tumor-directed radiation prior to immunotherapy upregulates programs of antitumor immune surveillance and dramatically enhance the efficacy of αPD-1 ICI therapy, leading to a high rate of durable cure, similar to observations in recent early-phase clinical trials[7,8,31,32].

### Sequencing tdRT Prior to PD-1 blockade enhances host antitumor immunity

To evaluate whether treatment order influences the therapeutic benefit of combination immunoradiotherapy, we compared outcomes in animals receiving αPD-1 prior to tdRT versus tdRT followed by αPD-1 (Fig. 4A). Tumor growth analysis revealed that sequencing tdRT prior to αPD-1 resulted in a significant improvement in tumor control compared to αPD-1 → tdRT or monotherapy (Fig. 4B, Supplementary Fig. 6A). To better understand how this treatment order enhances efficacy, we examined antigen-specific immune responses. We observed a marked increase in OVA-specific CD8 + T cells in tumors of animals treated with tdRT → αPD-1 compared to the reverse order, as assessed by tetramer staining (Fig. 4C; gating strategy in Supplementary Fig. 6B). In parallel, we detected an increased frequency of H-2Kb-SIINFEKL+ cross-presenting APCs in the sentinel lymph node (Fig. 4D), suggesting enhanced priming of antigen-specific T cells. These findings demonstrate that sequencing tdRT prior to αPD-1 therapy upregulates antigen presentation and primes host antitumor immune responses, thereby enhancing the efficacy of immune checkpoint blockade.

### Functional regional lymph nodes and intact immune cell trafficking are required for the host immune response to tumor-directed immunoradiotherapy

Having found that sequencing tdRT prior to αPD-1 drives tumor rejection through enhanced cross-presentation in the SLN and priming

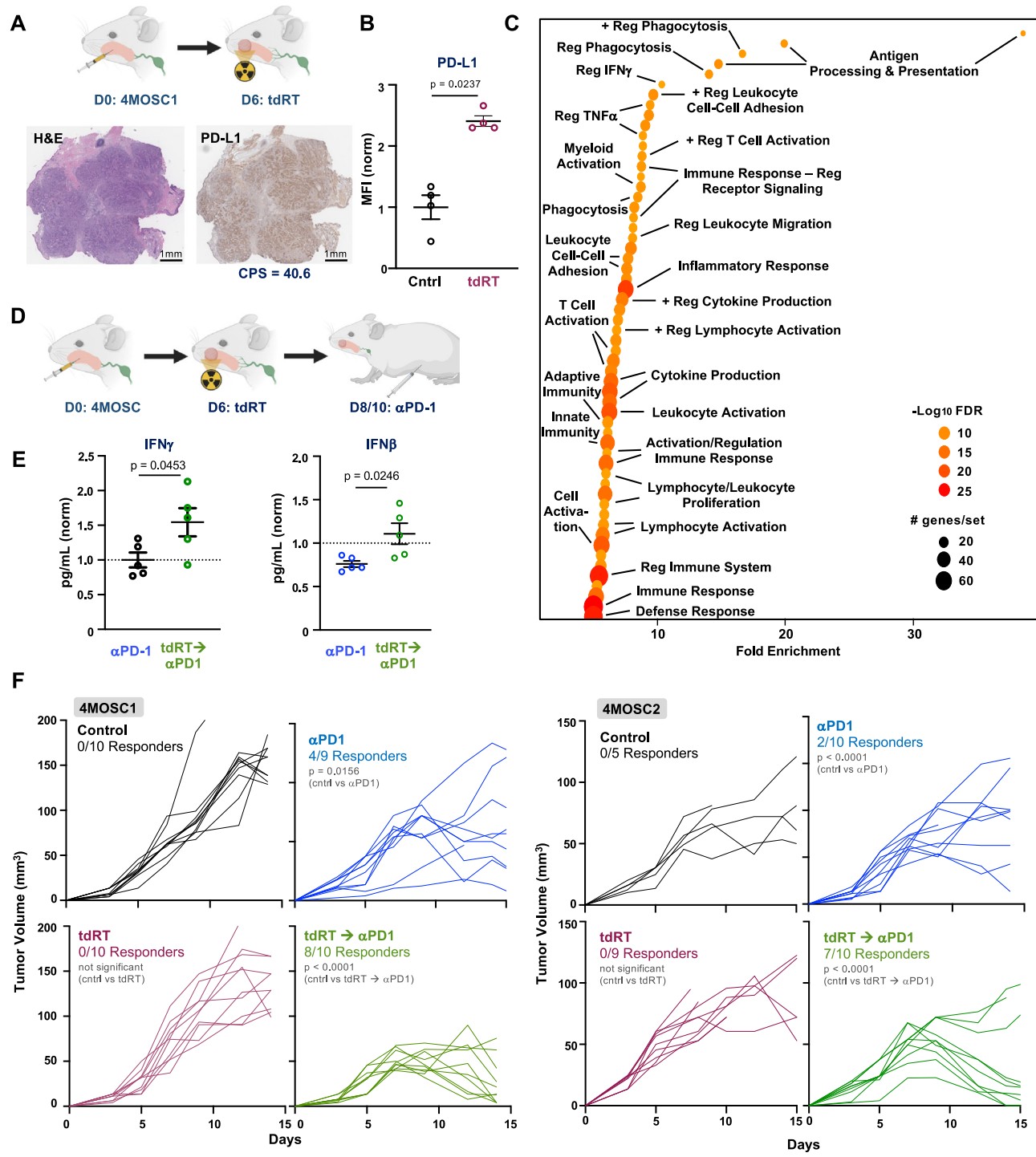

of antigen-specific CD8 + T cells (Fig. 4C, D), we next investigated whether the host response to sequenced tdRT→αPD-1 was dependent on the tumor-draining lymph node. WT animals bearing 4MOSC1 tumors were treated with various combinations of tumor-directed radiation and αPD-1 therapy in combination with therapies that ablate or impair regional lymphatic function: tdRT→αPD-1, elective nodal irradiation (ENI) + tdRT→αPD-1, and neck dissection (ND) + tdRT→αPD-1 (Fig. 4E). Tumor growth curves indicated significant tumor regression in the tdRT→αPD-1 group compared to controls and other treatment groups (Fig. 4F, left). Overall survival analysis demonstrated significant improvement in the tdRT→αPD-1 group, with a survival rate of 80% at 60 days (Fig. 4F, right).

Next, we examined the role of lymphatic trafficking in the antitumor response to tdRT→αPD-1. We found that tdRT→αPD-1 treatment led to increased levels of CXCL10 expression and significantly increased CD8 + T cell infiltration in the TME post-treatment (Fig. 4G, H). Sequestration of lymphocytes in secondary lymphoid organs with FTY720 (a sphingosine-1-phosphate receptor modulator) and or pharmacologic depletion of CD8 + T cells both blocked the tumor response to treatment, suggesting the critical role of lymphatic trafficking in mediating the antitumor effects of tdRT→αPD-1 (Fig. 4I, Supplementary Fig. 6C). These results highlight the requisite lymphatic migration in facilitating immunoradiotherapy efficacy.

**Fig. 3 | Tumor-directed radiation upregulates programs of antitumor immune surveillance to potentiate the αPD-1 ICI tumor response. A** (left) Cartoon schema of the experimental approach. WT animals injected with 4MOSC1 tumors were treated with tdRT on day 6. (right) Representative H&E and PD-L1 staining of tumor sections post-tdRT treatment. Combined Positive Score (CPS) for PD-L1 staining was 40.6. Tumor samples were fixed in zinc formalin fixative, embedded, sectioned, and stained. Immunohistochemistry for PD-L1 was performed using an anti-PD-L1 antibody, and CPS was calculated based on the ratio of PD-L1-positive cells to total viable tumor cells. Representative of $n = 4$ biologically independent samples; the experiment was independently repeated at least twice with similar results. **B** Quantification of PD-L1 expression on tumor cells post-tdRT, measured by flow cytometry, shown as normalized median fluorescence intensity (MFI) on live CD45− cells ($n = 4$ mice per group, $p = 0.0237$). Tumors were isolated, processed into single-cell suspensions, and stained with fluorochrome-conjugated anti-PD-L1 and CD45 antibodies. Flow cytometry was performed to assess PD-L1 expression levels. Data are presented as mean values ± SEM ($n = 4$ biologically independent samples/group); $p$ values calculated by two-sided unpaired Student's $t$ test. **C** Pathway enrichment analysis of immune-related gene expression changes in tumors following tdRT→αPD-1 ICI, highlighting significant upregulation in pathways involved in antigen processing and presentation, phagocytosis, and T cell activation. Tumors were harvested, RNA was isolated, and RNA sequencing was performed. Gene set enrichment analysis was conducted using GSEAPreranked module on the GenePattern public server. X-axis represents gene sets ranked by normalized enrichment score (NES); Y-axis represents the −$\log_{10}$(FDR q-value). **D** Cartoon schema of the experimental approach. WT animals injected with 4MOSC1 tumors were treated with tdRT on day 6 and subsequently treated with αPD-1 on day 8 and 10. **E** Cytokine analysis with normalized quantification of IFNγ and IFNβ levels shows significant increases in the tdRT→αPD-1 group compared to the αPD-1 only group ($n = 4$–5/group; $p = 0.0453$ for IFNγ, p = 0.0246 for IFNβ). Data are presented as mean values ± SEM ($n = 5$ biologically independent samples/group); $p$ values calculated by two-sided unpaired Student's t-test. **F** Tumor growth curves in 4MOSC1 and 4MOSC2-tumor bearing animals treated with αPD-1 or tdRT followed by αPD-1 (**left** and **right**, respectively). Left panels show control (0/10 responders) and αPD-1 treated groups (4/9 responders for 4MOSC1, $p = 0.0156$; 2/10 responders for 4MOSC2, $p = 0.0156$). Right panels show tdRT (0/10 responders) and tdRT followed by αPD-1 treated groups (8/10 responders for 4MOSC1, $p < 0.0001$; 7/10 responders for 4MOSC2, $p < 0.0001$). The data indicate that the combination therapy leads to a significantly higher response rate and tumor regression compared to αPD-1 alone. Best-fit lines and $p$ values calculated by simple linear regression (two-sided). Created in BioRender. Saddawi-Konefka, R. (2025) https://BioRender.com/m675tdd. Source data are provided as a Source Data file.

## Tumor-directed immunoradiotherapy reprograms dendritic cells to enhance antitumor surveillance and migration to the sentinel lymph node

The SLN plays a crucial role in host immunity, acting as the primary site for antigen presentation and immune cell activation. Dendritic cells within the SLN and tumor draining LN are pivotal for initiating and maintaining effective antitumor immune responses[34–36]. We aimed to elucidate how tdRT and sequenced therapy enhances dendritic cell surveillance and subsequent antitumor immunity by focusing on the SLN.

To investigate the effects of tdRT on the SLN, WT animals bearing 4MOSC1 tumors were treated and subjected to SLN mapping (Fig. 5A, Supplementary 7A–D). While bulk-RNA sequencing of the SLN in tumor-bearing animals following monotherapy with αPD-1 led to minimal significant changes (Supplementary Fig. 7A, B), analysis of SLN after tdRT→αPD-1 revealed significant upregulation of immune response pathways post-treatment, including those related to antigen processing, phagocytosis, and cell recognition, indicating enhanced immune activation (Fig. 5A, Supplementary Fig. 7C, D). ELISA quantification showed elevated levels of CCL19 in the TME post-treatment, suggesting increased chemokine production and recruitment of dendritic and other immune cells (Fig. 5B). Flow cytometric analysis confirmed a significant increase in activated CCR7+ dendritic cells (Fig. 5C) and activated H-2Kb-SIINFEKL+ cross-presenting dendritic cells (Supplementary Fig. 7E) in the SLN of treated mice.

We then characterized the tumor-SLN immune migratome in response to tdRT→αPD-1 compared to untreated animals. Similar to our approach in Fig. 1H, 4MOSC1-tumor bearing R26 x Ai9 animals, either treated with tdRT aPD1 or vehicle, were labeled with primary tumor injection of tamoxifen at day 10 followed by SLN mapping and harvesting for downstream CITE-sequencing of tdTomato+ sorted cells at day 13 (Fig. 5D, Supplementary. 5F). We identified distinct DC subpopulations, revealing a specific population with enhanced expression of key markers of DC activation - namely, CD40, CD86, and MHC II, indicating functional diversity (Fig. 5E, Supplementary. 5G). Multiplex immunofluorescence confirmed these findings, showing increased expression of DC markers in the SLN post tdRT treatment compared to untreated controls (Fig. 5F). Analysis of the average expression from the migratory DC-3 population revealed that tdRT→αPD-1 induced activation in pathways related to DC activation, phagocytosis, antigen processing, and interferon signaling, further supporting the enhanced immune surveillance role of DCs after tdRT→αPD-1 (Fig. 5G). We also performed CITE-seq analysis of tdTomato+ CD8 + T cells in SLNs following tdRT→αPD-1, which revealed increased expression of genes involved in activation, cytokine signaling, and interferon response (Supplementary Fig. 7H). These results suggest that tumor-targeting therapies can reprogram migratory dendritic cells from their typical pro-tolerogenic function towards an antitumor phenotype.

The tdRT→αPD-1 induced migratory CD86/CD40/MHC II/CCR7 + DC-3 population also exhibited relatively high expression of Batf3, the canonical transcription factor driving differentiation of type I conventional dendritic cells (cDC1s), cross-presenting DCs that are required within the tumor-draining lymph node during anticancer host response to immunotherapy[10,37]. Flow cytometric analysis showed a significant increase in cDC1 in the SLN post-treatment (Fig. 5H). To assess the requirement of cDC1s during tdRT→αPD-1, we employed Batf3−/− animals, which lack cDC1s. Tumor growth curves for Batf3−/− animals treated with tdRT→αPD-1 indicated that the absence of cDC1 abrogates the antitumor response, underscoring the critical role of these cells in mediating the therapeutic effects (Fig. 5I).

These findings collectively underscore that tumor-directed, sequenced immunoradiotherapy not only enhances the migratory potential of CD86/CD40/MHCII/CD11c+ dendritic cells to the SLN but, more importantly, supports a model in which these cells undergo functional reprogramming. While CCR7 expression is characteristic of migratory dendritic cells, post-treatment, these cells exhibit enhanced expression of additional activation markers, such as CD86 and CD40, which are crucial for effective T-cell priming and antitumor responses. This reprogramming is further supported by upregulated IFN signaling pathways, which are integral to the amplification of immune responses and the activation of cytotoxic T cells. These results emphasize the pivotal role of this reprogrammed dendritic cell population in orchestrating effective antitumor immunity and underscore the necessity of preserving and targeting these cells to optimize immunotherapeutic strategies.

## An intact sentinel node lymphatic channel and migration of activated dendritic cells is required for effective immunoradiotherapy and tumor-directed T cell response

Given the crucial role of tumor-sentinel node migrating dendritic cells in facilitating immune responses, we aimed to determine the specific necessity of their trafficking to the sentinel lymph node.

To assess the impact of disrupting the sentinel lymphatic channel on dendritic cell migration, we performed lymphatic channel ablation (LCA) on the SLN or non-sentinel lymph nodes (nSLN) in 4MOSC1

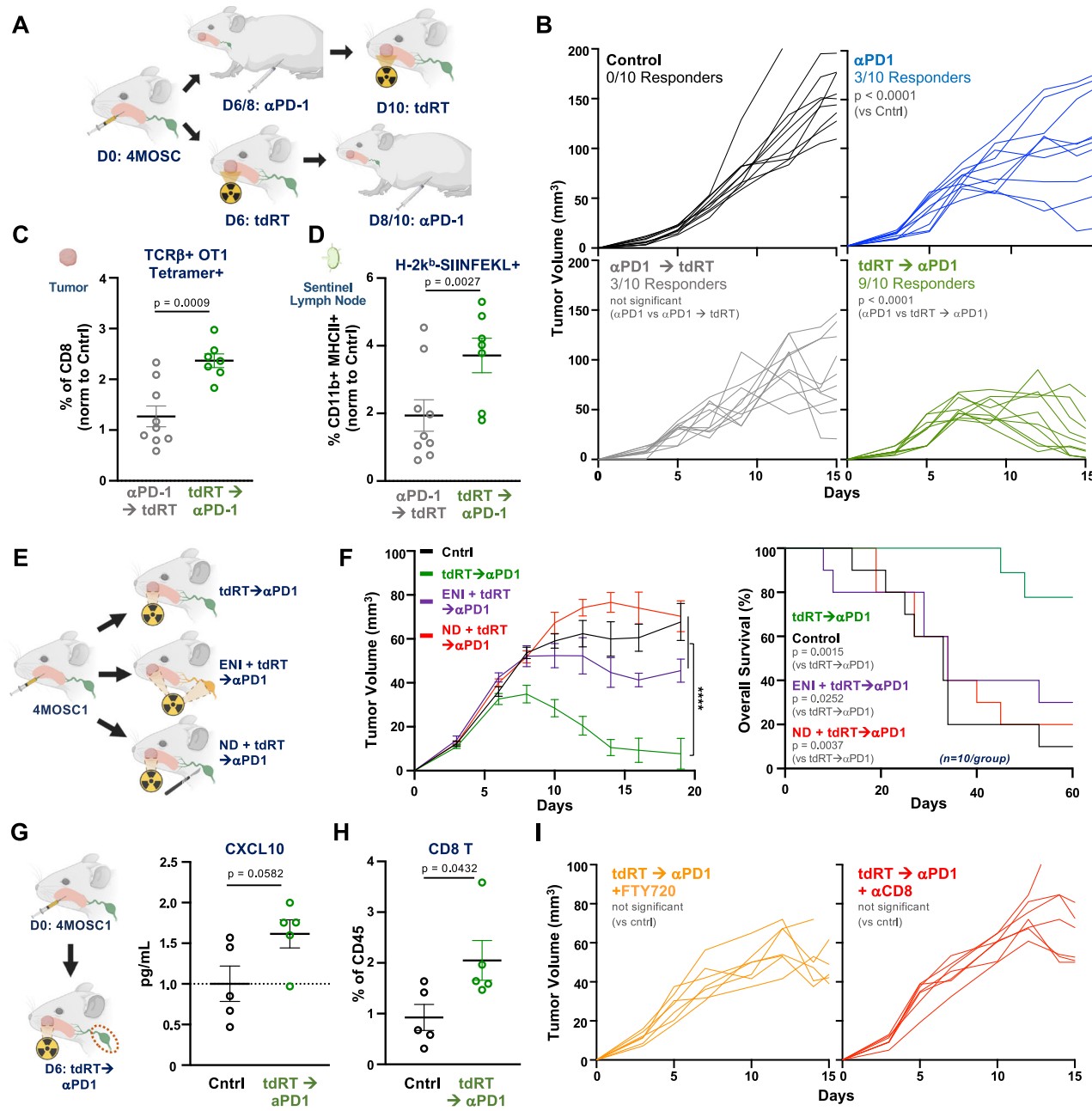

tumor-bearing mice, followed by tdRT→αPD-1 treatment (Fig. 6A, Supplementary Fig. 8A, B). Flow cytometric analysis showed a significant reduction in the percentage of dendritic cells in the SLN post-treatment in SLN LCA mice compared to controls (Fig. 6B). Additionally, flow cytometric analysis of Ovalbumin-specific T cells in the SLN indicated a marked decrease in T cell activation in SLN LCA mice (Fig. 6C, Supplementary Fig. 8C). Tumor growth curves demonstrated that SLN LCA significantly impaired tumor regression in response to tdRT→αPD-1, whereas nSLN LCA did not affect the treatment outcome (Fig. 6D).

Previous studies have shown that CCR7+ DCs exclusively utilize MMP9 to enter lymph nodes, a key step in mounting an effective immune response[38–40]. We further investigated the role of MMP9 in facilitating dendritic cell migration in 4MOSC1 tumor-bearing mice treated with tdRT→αPD-1 (Fig. 6E). Direct intratumoral injection of MMP9 inhibitor prior to and during tdRT→αPD-1 treatment significantly reduced CCR7 expression on dendritic cells in SLN (Fig. 6F) and decreased the number of CCR7 + MHCII+ dendritic cells in the SLN

(Fig. 6F). Correspondingly, tumor growth curves showed that MMP9 inhibition abrogated the antitumor efficacy of tdRT→αPD-1 (Fig. 6G).

Finally, we examined the effect of physical interruption of cell migration on the development of an effective T cell response in response to tdRT→αPD-1. TCR analysis showed a significant reduction in the productive frequency and clonality of T cell responses in the SLN and tumor of SLN LCA mice compared to tdRT→αPD-1 alone (Fig. 6H). This was further supported by TCR clonal analysis, which revealed decreased enrichment of specific CDR3 sequences in SLN LCA mice (Fig. 6I). These findings underscore the critical importance of intact sentinel lymphatic channels and MMP9-dependent entry of CCR7+ dendritic cells into the SLN for the efficacy of tumor-directed immunoradiotherapy.

## Discussion

Incorporation of immunotherapy into treatment paradigms for locally advanced HNSCC remains a challenge, particularly given the consistent lack of effect with the addition of concurrent immunotherapy

**Fig. 4 | The host response to sequenced, tumor-directed immunoradiotherapy is coordinated in regional lymphatics. A** Cartoon schema of the experimental approach. WT animals bearing 4MOSC1 tumors were treated with αPD-1 on days 6 and 8 (αPD-1 alone); αPD-1 on days 6 and 8 followed by tdRT on day 10 (αPD-1 → tdRT); or tdRT on day 6 followed by αPD-1 on days 8 and 10 (tdRT→αPD-1). **B** Tumor growth curves from 4MOSC1-tumor bearing animals treated with αPD-1 alone, αPD-1 followed by tdRT (αPD-1 → tdRT) or tdRT followed by αPD-1 (tdRT → αPD-1). tdRT → αPD-1 sequence demonstrated significantly enhanced tumor control compared to αPD-1 alone and other sequencing strategies ($n = 10$ mice per group, p < 0.0001 for tdRT → αPD-1 vs αPD-1). Best-fit lines and p values calculated by simple linear regression (two-sided). **C** Quantification of tumor-infiltrating, antigen-specific CD8 + T cells (TCRβ + OT1 Tetramer +) across treatment groups. tdRT → αPD-1 therapy significantly increased the proportion of tetramer+ CD8 + T cells compared to αPD-1 → tdRT ($n = 9$ mice per group, $p = 0.0008$). Data are presented as mean values ± SEM; p values calculated by two-sided unpaired Student's t test. **D** Quantification of cross-presenting APCs (H-2Kb-SIINFEKL +) in sentinel lymph nodes across treatment groups. tdRT → αPD-1 therapy significantly increased SIIN-FEKL presentation compared to αPD-1 → tdRT ($n = 9$ mice per group, $p = 0.0027$). Data are presented as mean values ± SEM; p values calculated by two-sided unpaired Student's t test. **E** (left) Cartoon schema of the experimental approach. WT animals injected with 4MOSC1 tumors were treated with various combinations of tumor-directed radiation and αPD-1 therapy: tdRT→αPD-1, elective nodal irradiation (ENI) + tdRT→αPD-1, and neck dissection (ND) + tdRT→αPD-1. Tumor-directed radiation therapy was delivered using the SmART-Plan system with a 5 mm collimator, and a single fraction of 4 Gy directed to the tumor or tumor and draining lymphatics (ENI)

was administered as described in the methods. **F** (left) Tumor growth curves for control and different treatment groups (tdRT→αPD-1, ENI + tdRT→αPD-1, ND + tdRT→αPD-1) ($n = 10$ mice per group, p < 0.0001). (right) Overall survival curves for the same treatment groups showing significant differences ($p = 0.0015$ for tdRT→αPD-1 vs control, $p = 0.0252$ for ENI + tdRT→αPD-1 vs tdRT→αPD-1, $p = 0.0037$ for ND + tdRT→αPD-1 vs tdRT→αPD-1). Data are presented as mean values ± SEM. Best-fit lines and p values calculated by simple linear regression (two-sided). **G** (left) Cartoon schema of the experimental approach for measuring CXCL10 levels. WT animals injected with 4MOSC1 tumors were treated with tdRT→αPD-1. (right) Quantification of CXCL10 levels in the TME post-treatment ($n = 5$ mice per group, $p = 0.0582$). CXCL10 levels were measured using a mouse chemokine array on tissue homogenates from treated tumors. Data are presented as mean values ± SEM; p values calculated by two-sided unpaired Student's t test. **H** Flow cytometric analysis of CD8 T cell infiltration in the TME post-treatment ($n = 5$ mice per group, $p = 0.0432$). Tumors were processed into single-cell suspensions, stained with fluorochrome-conjugated anti-CD8 antibodies, and analyzed by flow cytometry. Data are presented as mean values ± SEM; p values calculated by two-sided unpaired Student's t test. **I** Tumor growth curves for tdRT→αPD-1 combined with FTY720 (left) and αCD8 (right) ($n = 7$ mice per group, not significant). FTY720 (Cayman 10006292) 5 mg/kg/day IP was administered to inhibit lymphocyte egress from lymph nodes, and αCD8 antibodies (BioXCell, clone YTS169.4) 250 mg/mouse/dose IP were used for CD8 depletion every other day. Best-fit lines and p values calculated by simple linear regression (two-sided). Created in BioRender. Saddawi-Konefka, R. (2025) https://BioRender.com/m675tdd. Source data are provided as a Source Data file.

to traditional broad lymphatic radiation and concurrent chemotherapy[3–6,10]. Therapeutic strategies that spare draining lymphatic basins during delivery of immunotherapy show promising tumor responses, suggesting that preserving lymphatic function is crucial for an effective host immunotherapeutic response[41]. Subsequent efforts reinforce this premise by demonstrating the critical role of tumor-draining lymph nodes in mediating host response to immunotherapy[7,8,32,42,43]. Here, we detail a strategy combining lymphatic-preserving, tumor-directed radiotherapy followed by immune checkpoint inhibition, revealing how this sequenced strategy enhances immune surveillance and orchestrates a robust antitumor response. Our analysis of the tumor-sentinel lymph node immunomigratome reveals a distinct and diverse immune cell repertoire, highlighting the unique role of the SLN in orchestrating effective antitumor immunity. Specifically, our findings highlight the pivotal role of migratory dendritic cells in facilitating effective sequenced radiation and immunotherapy through cell trafficking to sentinel lymph nodes, underscoring the importance of optimizing DC migration to enhance therapeutic outcomes. Ultimately, by rationally sequencing immune-sensitizing, tumor-directed radiotherapy followed by checkpoint inhibition, we characterize the tumor-sentinel lymph node immunomigratome that drives durable antitumor responses and provide a rational basis for application of this therapeutic approach in HNSCC and other systems.

Recent clinical experience demonstrates that sequencing neoadjuvant tumor-directed radiotherapy with subsequent checkpoint blockade immunotherapy can significantly enhance host antitumor immunity and tumor response. For instance, a phase 2 trial in early-stage non-small-cell lung cancer showed that neoadjuvant durvalumab sequenced after tumor-targeted stereotactic body radiotherapy (8 Gy × 3 fractions) improved major pathological response rates from 6.7% to 53.3%, highlighting a potent immunomodulatory effect of neoadjuvant SBRT[31,32]. In HNSCC, early-phase trials combining hypofractionated stereotactic body radiation therapy with durvalumab or nivolumab demonstrated high rates of major pathological response and overall survival, further supporting the efficacy of this approach[7,8]. Importantly, the Leidner study used a "sandwich" design where radiotherapy was administered between courses of nivolumab, highlighting the implications of sequencing of therapies. Secondary endpoint analyses in these studies demonstrate that immune checkpoint

inhibition, when sequenced with tumor-targeted immunomodulation, significantly enhances host antitumor immunity by increasing effector T cells and decreasing immunosuppressive populations in peripheral blood[7], as well as enhancing programs of antigen presentation[8]. While these early-phase studies support lymphatic-sparing strategies, it is important to interpret their results with caution. For instance, the phase II study by Ma et al. investigating neoadjuvant tdRT combined with anti-PD-L1 and anti-CTLA-4 in HPV + HNSCC demonstrated impressive immune priming and tumor response; however, a 26% locoregional recurrence rate was observed at a median of 3 months following pathologic complete response[9]. These findings underscore the need to better understand how lymphatic structures contribute to durable immunity and to refine the selection, timing, and scope of radiation therapy in immunotherapy-based regimens. Collectively, these results indicate that achieving lymphatic sparing by restricting radiotherapy dosage to the tumor volume and sequencing it appropriately with immunotherapy can potentiate immunotherapy and significantly improve clinical outcomes. Understanding the biological underpinnings of these clinical observations in animal models is crucial for optimizing future treatment strategies and therapy designs.

The dynamics of how the host antitumor response is coordinated across the tumor-lymphatic axis during sequenced neoadjuvant IO have not been directly addressed. In this study, we explore the mechanisms underlying immune cell trafficking and activation within the tumor-sentinel lymph node axis that can inform the development of effective immune therapies aimed at achieving durable antitumor responses. First, we map the sentinel lymph node in orthotopic tumor-bearing animals and develop a spatiotemporal reporter model system to characterize locoregional trafficking of immune cells at single-cell resolution, defining the "cancer immunomigratome" from the primary tumor to the sentinel lymph node. To the best of our knowledge, our work represents the first report of the cancer immunomigratome and its diversity, revealing an immunologic niche worthy of further investigation. This not only characterizes the distinct and dynamic immune cell populations migrating to the sentinel lymph node from the TME but also highlights their critical roles in orchestrating antitumor immunity. By elucidating these migratory pathways and cellular interactions, our findings provide a foundational framework for future research aimed at optimizing immunotherapeutic strategies and improving clinical outcomes in cancer treatment. Next, we develop a

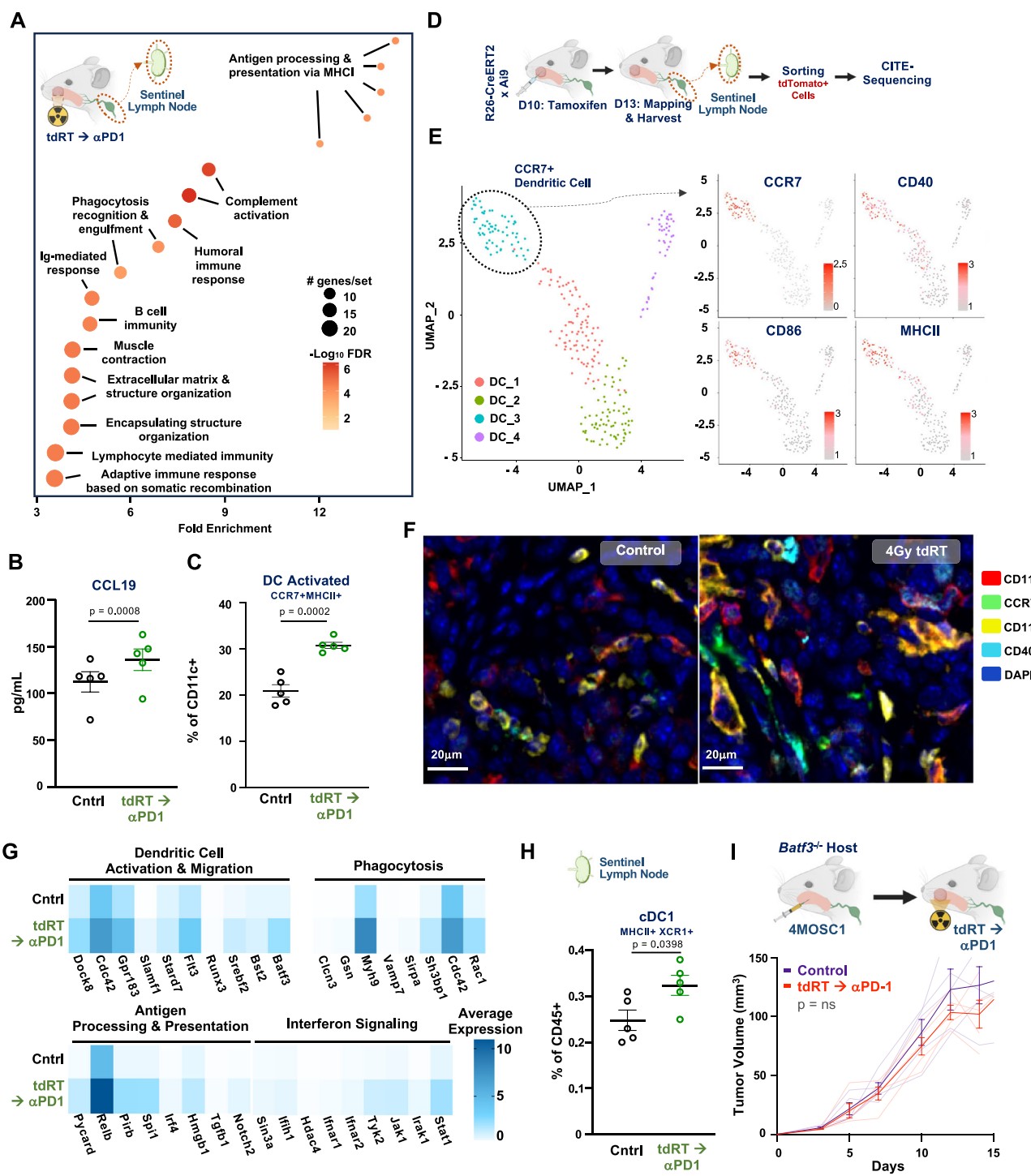

model of tumor-directed immunomodulatory therapy—specifically, sub-cytotoxic lymphatic sparing radiotherapy to gross tumor volume—which modulates the tumor immune microenvironment without inducing tumor regression. Our findings indicate that tumor-directed, lymphatic sparing radiotherapy potentiates the response to aPD-1 ICI, achieving complete and durable responses. By analyzing the immunomigratome during tumor-directed immunoradiotherapy, we identify enhanced programs of antitumor immunosurveillance, specifically activated migratory dendritic cells. Ultimately, we find that either physical interruption of the lymphatic channel to the SLN or selective blockade of CCR7+ migratory dendritic cells' entry into the SLN abrogates the response to sequenced immunoradiotherapy and the expansion of the T cell repertoire. Mechanistically, our findings

support the concept of lymphatic-preserving immunotherapeutic strategies by elucidating the biology of immune surveillance from the primary tumor to the SLN.

The SLN is increasingly recognized as a key immunologic hub, playing an active role in orchestrating and coordinating the antitumor immune response[16,17]. SLNs are sites of immune modulation, where DCs and other antigen-presenting cells interact with T cells, potentially enhancing or dampening the immune response depending on the context. The role of draining lymph nodes in response to ICIs has significant parallels to host response to infection[44-47]. For example, Leal et al. showed that lymph nodes efficiently capture and present bacterial antigens, leading to T cell activation and robust immune cell recruitment[48]. Similarly, Pirillo et al. highlighted lymph nodes as

**Fig. 5 | Tumor-directed immunoradiotherapy enhances dendritic cell anti-tumor surveillance across the tumor and sentinel lymph node. A** Cartoon schema of the experimental approach. WT animals injected with 4MOSC1 tumors were treated with tdRT→αPD-1 and then subjected to sentinel lymph node (SLN) mapping. RNA sequencing from the sentinel lymph node showing normalized enrichment scores for various immune response pathways post-treatment. X axis represents gene sets ranked by normalized enrichment score (NES); Y-axis represents the $-\log_{10}$(FDR q value). **B** ELISA quantification of CCL19 levels in the TME post-treatment ($n = 5$ mice per group, $p = 0.0008$). Data are presented as mean values ± SEM ($n = 5$ biologically independent samples/group); p values calculated by two-sided unpaired Student's t test. **C** Flow cytometric analysis of activated dendritic cells (MHCII+ CD11c + CCR7 +) in the SLN post-treatment ($n = 5$ mice per group, $p = 0.0432$). Data are presented as mean values ± SEM; p values calculated by two-sided unpaired Student's t test. **D** Cartoon schema of the experimental approach. ROSA-26 x Ai9 animals were injected with 4MOSC1 tumors were treated with tdRT→αPD-1, labeled with tamoxifen in the tumor and then subjected to sentinel lymph node (SLN) mapping. Sorted live tdTomato+ cells from the SLN were then sent for CITE-sequencing. CITE-sequencing was performed on sorted tdTomato+ cells isolated from the SLNs, as described in the methods. $n = 2$ biologically independent samples/group. **E** (left) Subsampled dendritic cell populations after CITE-sequencing, showing UMAP plots with clusters of dendritic cells (right) Seurat

objects featuring the expression of CCR7, CD40, CD86, and MHCII across DC populations. UMAP clustering and Seurat object analysis were conducted on the CITE-sequencing data to identify dendritic cell subpopulations. **F** Multiplex immunofluorescence from sentinel lymph nodes showing CD11c, CCR7, CD11b, CD40, and DAPI staining in control and tdRT-treated groups. Representative of $n = 3$ biologically independent samples; experiment was independently repeated at least twice with similar results. **G** Analysis of average expression from the DC-3 population showing activation in programs of DC migration, phagocytosis, antigen processing and presentation, and interferon signaling pathways. Representative of $n = 2$ biologically independent samples. **H** Flow cytometric analysis of conventional dendritic cells (cDC1, MHCII + XCR1 +) in the SLN post-treatment ($n = 5$ mice per group, $p = 0.0398$). Data are presented as mean values ± SEM; p values calculated by two-sided unpaired Student's t test. **I** (left) Cartoon schema of the experimental approach for Batf3−/− animals, which lack conventional type I dendritic cells. (right) Tumor growth curves for control and tdRT→αPD-1 treatment in Batf3−/− animals ($n = 5$ mice per group, $p = $ ns). Data are presented as mean values ± SEM ($n = 5$–6 biologically independent samples/group). Best-fit lines and p values calculated by simple linear regression (two-sided). Created in BioRender. Saddawi-Konefka, R. (2025) https://BioRender.com/m675tdd. Source data are provided as a Source Data file.

critical sites for antigen presentation by migratory dendritic cells during viral infections and tumor responses[49]. Our findings here reveal a parallel role for the SLN in mounting an effective antitumor response. Specifically, we demonstrate that activated, migratory CCR7+ dendritic cells are requisite for the complete tumor response to immunoradiotherapy.

DCs play a critical role in the initiation and regulation of immune responses by capturing antigens at peripheral sites and migrating to draining lymph nodes. In homeostasis, the role of migratory DCs has been described as pro-tolerogenic and immunosuppressive, with DCs trafficking from the periphery to prime regulatory T cells against self-antigens in draining lymph nodes[50]. During neoplasia, however, these migratory cells can acquire an antitumor role by initiating an adaptive immune response[10,40,51]. Specifically, the migratory CCR7 + DC, guided to the draining lymph node by its exclusive endogenous ligands CCL19 and CCL21, has been characterized as pivotal in this process[52]. Aligned with these reports, our immunomigratome analysis has revealed that tumor-sentinel node migratory CD86/CD40/MHC II/CCR7+ DCs are enhanced by IRT, and CCR7 + DC are essential for the expansion and activation of tumor-specific T cells within the sentinel lymph node to mediate a successful tumor response to immunoradiotherapy. Work investigating the role of migratory DCs in mediating rheumatologic diseases has described the obligate role for matrix metalloproteinase-9 in the trafficking of migratory DCs, where MMP9 serves as the critical enzyme facilitating CCR7+ DCs' entry into draining lymph nodes. Specifically, inhibition of MMP-9 significantly impairs the migration of DCs, reducing their ability to initiate immune responses[38]. MMP-9-deficient DCs show markedly reduced migration through tissues, underscoring the enzyme's role in DC trafficking[40]. These findings suggest that MMP-9 is indispensable for the effective migration and function of CCR7+ DCs, thereby influencing the overall success of immunotherapeutic strategies, which may explain the otherwise puzzling failure of clinical trials examining MMP9i in cancer[53]. Together, these insights highlight the critical role of CCR7+ and other migratory DCs in antitumor immunity and facilitating their migration, consistent with observations that enhancing DC function and migration may be achieved by immune adjuvants, such as TLR9 agonists or anti-CD47, or through the use of DC vaccines[54,55].

Our findings are consistent with existing literature indicating that radiation therapy plays a crucial role in modulating the immune system, thereby enhancing the efficacy of immunotherapy. Traditionally employed for its cytotoxic effects, radiotherapy is now appreciated for its ability to influence antitumor immunity through two primary

mechanisms: radiation-induced immunogenic cell death and radiation immunomodulation. Radiation-induced immunogenic cell death involves several key molecular signals, such as calreticulin exposure, ATP release, and HMGB1 release, which collectively lead to the activation and maturation of dendritic cells and initiate a robust antitumor immune response[21]. Beyond inducing cell death, radiation modulates the tumor-immune microenvironment by driving programs of antigen sampling and antigen-presenting cell surveillance[24,28,56]. Furthermore, sublethal radiation doses, as employed in this study, can alter the phenotype of tumor cells, making them more susceptible to immune attack by upregulating death receptors, costimulatory molecules, and stress-induced ligands[21], and influences activation pathways typically targeted by agonism of damage-associated molecular patterns or pathogen-associated molecular patterns pathways, as well as dendritic cell vaccines. By modulating the immune landscape within the TME, immunosensitizing radiation therapy can amplify the host immune response against cancer, synergistically enhancing the efficacy of immunotherapy and paving the way for therapeutic strategies. As noted, the timing and order of these therapies are crucial in maximizing their antitumor potential, and additional investigation is needed to understand the optimal dosing and timing to fully leverage the benefits of targeted radiation as an immunosensitizer.

Our work here provides evidence for leveraging lymphatic sparing, tumor-directed radiotherapy to enhance immunotherapeutic response, but it does have limitations. Although our model faithfully mimics the human-disease counterpart and accurately represents many aspects of the host immune response, including genetic and treatment-related variables, there are inherent differences between murine and human systems that must be acknowledged. Although we find that sub-cytotoxic and targeted radiotherapy can serve as a potent immunomodulator to potentiate ICI, harmonizing these insights within current radiotherapy treatment paradigms is not straightforward. Cytotoxic radiotherapy is a longstanding clinical care standard that may or may not have equivalent immunostimulatory effects; murine radiotherapy dosing is not directly translatable, and the TME can differ significantly between species. Additionally, the integration of surgical sentinel node mapping, neck dissection, or other clinical interventions with treatment paradigms provides challenges. Furthermore, while our model highlights the key role of the sentinel lymph node, it is essential to consider the potential role of other lymph nodes in immune surveillance and response, and lymphatic drainage may differ between human and murine systems. A key avenue for further investigation includes the plasticity of the immune response in

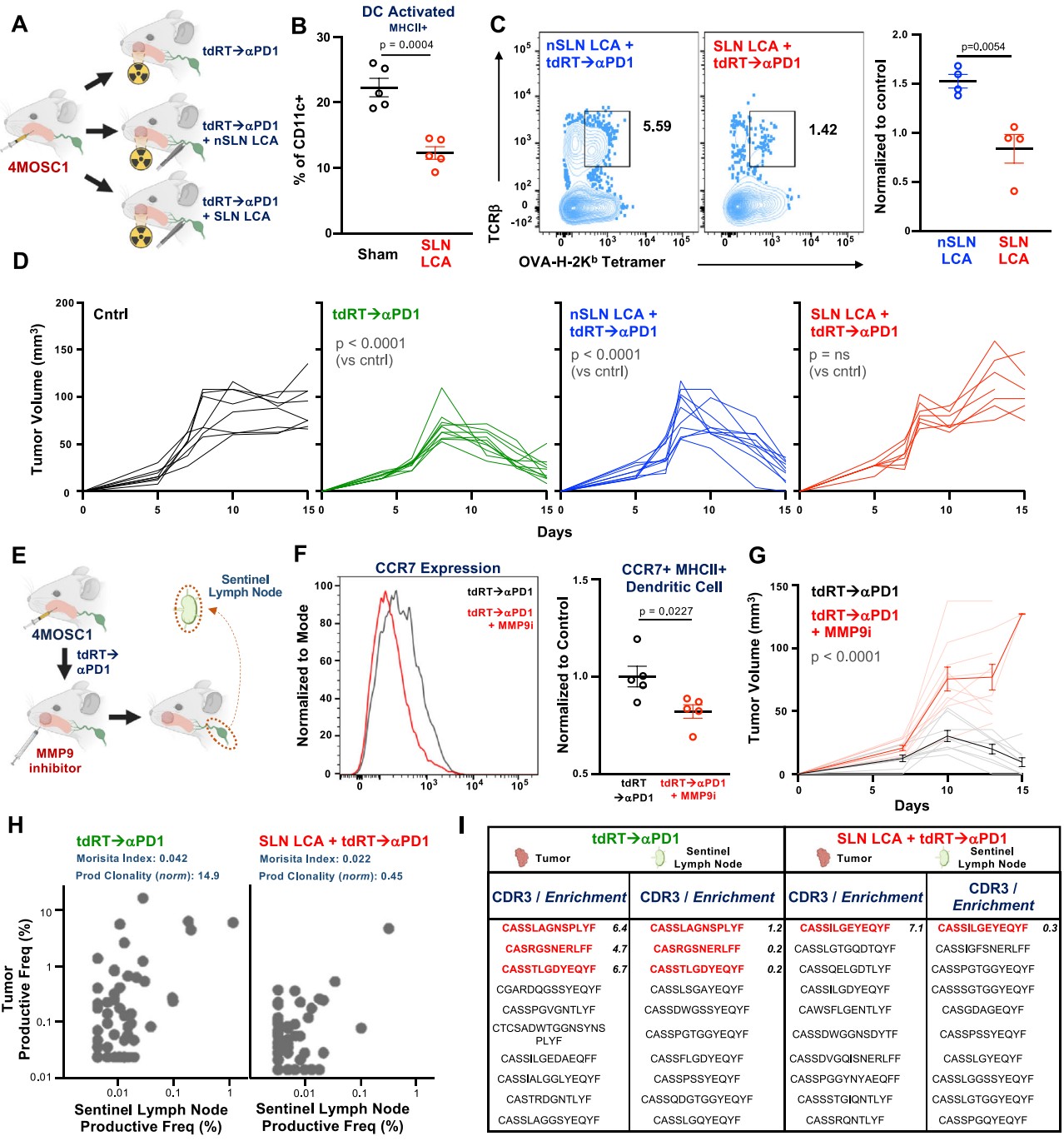

the context of the SLN, adjacent lymph nodes, and other draining and nondraining lymphatic basins and investigate the feasibility of lymphatic-sparing approaches in other systems. Characterizing DC function and enhancing DC migration through targeted therapies, such as SLN-specific interventions, could provide further insights into optimizing immunotherapy outcomes. It is clear that in addition to DCs, there is a robust complement of immune cells migrating from the tumor to the SLN that are key to mounting a tumor immune response. Further characterization of the full complement and function of specific components of the tumor-SLN migratome and dynamic interactions among migrating cells is likely to provide further insights into the biology of tumor-SLN immune response. Together, these findings define the sentinel lymph node as a critical site of immune activation and control, revealing a coordinated tumor–lymphatic–immune axis that can be leveraged to guide next-generation immunotherapy.

## Methods

All the animal studies were approved by the University of California San Diego (UCSD) Institutional Animal Care and Use Committee (IACUC, protocol #S16200); all experiments adhere with all relevant ethical regulations for animal testing and research.

### Study design

ARRIVE 2.0 guidelines[57] for reporting animal research were employed as follows:

**Sample size**. The sample size for each experiment was selected in accordance with historical data from the preclinical models employed to achieve significance; described in detail in each experiment and the statistical analysis section below.

**Fig. 6 | CCR7+ dendritic cell trafficking and MMP9-dependent entry into the sentinel lymph node are critical for immunoradiotherapy efficacy. A** (left) Cartoon schema of the experimental approach. WT animals injected with 4MOSC1 tumors were treated with tdRT→αPD-1 with or without sentinel lymph node lymphatic channel ablation (SLN LCA) or non-sentinel lymph node lymphatic channel ablation (nSLN LCA). SLN LCA and nSLN LCA procedures were performed as described in the methods, involving precise surgical ablation of lymphatic channels. **B** Flow cytometric analysis of dendritic cell percentages in the sentinel lymph node post-treatment ($n = 5$ mice per group, $p = 0.0379$). Data are presented as mean values ± SEM; $p$ values calculated by two-sided unpaired Student's $t$ test. **C** Flow cytometric analysis of Ovalbumin-specific T cells (4MOSC1-LucOS model) in SLN and nSLN with and without SLN LCA (left). Quantification of normalized percentages relative to control (right) ($n = 4$ mice per group, $p = 0.0054$). Data are presented as mean values ± SEM; $p$ values calculated by two-sided unpaired Student's $t$ test. **D** Tumor growth curves for control, tdRT→αPD-1, nSLN LCA + tdRT→αPD-1, and SLN LCA + tdRT→αPD-1 groups ($n = 10$ mice per group, $p < 0.0001$). Best-fit lines and $p$ values calculated by simple linear regression (two-sided). **E** (left) Cartoon schema of the experimental approach. WT animals injected with 4MOSC1 tumors were treated with tdRT→αPD-1 and MMP9 inhibitor. (right) CCR7 expression on dendritic cells and quantification of normalized percentages relative to control ($n = 5$ mice per group, $p = 0.0022$). MMP9 inhibition was achieved using a specific inhibitor (Sigma 444293) at 0.4 mg/mouse/dose delivered intratumorally in 8 μL volume on post-transplant day 3, 5 and 7, as previously described[38,40]. SLN were mapped and harvested 48 hours after completion of treatment. **F** Flow cytometric analysis of CCR7 + MHCII+ dendritic cells in the SLN post-treatment with and without MMP9 inhibition. Quantification of normalized percentages relative to control ($n = 5$ mice per group, $p < 0.0001$). Data are presented as mean values ± SEM; $p$ values calculated by two-sided unpaired Student's $t$ test. **G** Tumor growth curves for tdRT→αPD-1 with and without MMP9 inhibition ($n = 10$ mice per group, $p < 0.0001$). Data are presented as mean values ± SEM. Best-fit lines and $p$ values calculated by simple linear regression (two-sided). **H** TCR analysis showing productive frequency in the sentinel lymph node and tumor for tdRT→αPD-1 (Morisita Index: 0.042, Prod Clonality (norm): 14.9) and SLN LCA + tdRT→αPD-1 (Morisita Index: 0.022, Prod Clonality (norm): 0.45) groups. **I** TCR clonal analysis showing enriched CDR3 sequences in the sentinel lymph node and tumor for tdRT→αPD-1 and SLN LCA + tdRT→αPD-1 groups. Enriched CDR3 sequences were identified through TCR sequencing and analyzed for clonal distribution. Created in BioRender. Saddawi-Konefka, R. (2025) https://BioRender.com/m675tdd. Source data are provided as a Source Data file.

**Rules for stopping data collection.** In the case of in vivo experiments, stopping rules were pre-approved according to the University of California San Diego (UCSD) Institutional Animal Care and Use Committee (IACUC), with protocol ASP #S16200 (described below).

**Data inclusion/exclusion.** All data collected were included and represented in the main figures or supplementary materials.

**Outliers.** Outliers were included in the reported data.

**Replicates.** All experiments, when feasible, were repeated at least twice and reproducibility confirmed; in all possible instances, data from repeat experiments are represented.

**Research objectives.** The research objective did not alter and is as follows: to provide a mechanistic understanding of how standard oncologic therapies targeting regional lymphatics impact the tumor response to immune-oncology therapy, to define rational treatment sequences that mobilize systemic antitumor immunity, achieve optimal tumor responses, confer durable antitumor immunity, and control regional metastatic disease.

**Research subjects.** We employed translational preclinical models of HNSCC, as described below.

**Experimental design.** This work represents a controlled, laboratory investigation involving preclinical models of HNSCC. Treatments applied were designed to deliberately model current clinical therapies for HNSCC patients. In general, endpoints for studies presented include tumor growth kinetics, survival analyses and a spectrum of immunological analyses.

**Randomization.** All in vivo experiments were randomized by tumor volume prior to initiation of treatment or data collection. Source data are provided with this paper and have been added to a public repository, as specified in each experiment.

**Reagents**
In vivo antibodies were purchased from Bio X Cell (West Lebanon, NH). Fluorochrome-conjugated antibodies were purchased from BD Biosciences (San Jose, CA) and BioLegend (San Diego, CA). FTY720 (Cayman 10006292). All other chemicals and reagents were from Sigma-Aldrich (St. Louis, MO) unless indicated.

**Cell lines and tissue culture**
The 4MOSC1 syngeneic mouse HNSCC cells harboring a human tobacco-related mutanome and genomic landscape were developed and described for use in immunotherapy studies in our prior report[10,30]. MOC1 syngeneic mouse HNSCC cells derived from DMBA-induced oral tumors were generously provided by Dr. R. Uppaluri[58]. 293 T cells (ATCC CRL-3216) were cultured in Dulbecco's Modified Eagle's Medium (DMEM) supplemented with 10% fetal bovine serum, 2 mM L-glutamine (ATCC 30-2214) and 1% antibiotic/antimitotic solution. 4MOSC1 cells were cultured in Defined Ketatinocyte-SFM medium supplemented with EGF Recombinant Mouse Protein (5 ng/ml), Cholera Toxin (50 pM) and 1% antibiotic/antimycotic solution. MOC1 cells were cultured in HyCloneTM Iscove's Modified Dulbecco's Medium (IMDM; GE Healthcare Life sciences, South Logan, UT, USA, #sh30228.02)/HyCloneTM Ham's Nutrient Mixture F12 (GE Healthcare Life sciences# sh30026.01) at a 2:1 mixture with 5% fetal bovine serum, 1% antibiotic/antimycotic solution, 5 ng/mL EGF, 400 ng/mL hydrocortisone (Sigma-Aldrich, St Louis, MO, USA, #H0135), and 5 mg/mL insulin (Sigma-Aldrich, #I6634). All cells were cultured at 37 °C in the presence of 5% $CO_2$.

**TIL isolation and flow cytometry**
Tumors were isolated, minced, and resuspended in the Tumor Dissociation Kit (Miltenyi Biotec, San Diego, CA) diluted into DMEM for subsequent processing with the gentleMACS Octo Dissociator, according to the manufacturer's recommendations for tumor dissociation into a single-cell suspension. Digested tissues were then passed through 70-μm strainers to produce a single-cell suspension. Samples were washed with PBS and processed for live/dead cell discrimination using Zombie viability stains (Biolegend, San Diego, CA). Cell sus- suspensions were then washed with cell staining buffer (Biolegend 420201) prior to cell surface staining, performed at the indicated antibody dilutions for 30 min at 4 °C, and protected from light. Cell surface staining was performed for 30 min at 4 °C with the following mouse antibodies: CD45 (30-F11) (1:100), CD3 (17A2) (1:200), CD8a (53- 6.7) (1:100), CD4 (RM4-4) (1:100), Slamf6 (330AJ) (1:100), PD-1 (29 F.1A12) (1:100), CD44 (IM7) (1:100), CD19 (6D5) (1:100), CXCR3 (S18001A) (1:100), Tim3 (RMT3-23) (1:100), NK1.1 (PK136) (1:100), CD69 (H1.2F3) (1:100), CD62L (MEL- 14) (1:100), BST2 (129C1) (1:100), Ly6C (HK1.4) (1:100), CD11b (M1/70) (1:100), CD11c (N418) (1:100), Siglec H (551) (1:100), XCR1 (ZET) (1:100), CD64 (X54-5/7.1) (1:100), CD103 (2E7) (1:100), SIRPa (P84) (1:100), MHCII (M5/114.15.2) (1:200), CD80 (16-10A1) (1:100), CD86 (GL-1) (1:100), Ep-CAM (G8.8) (1:100) and H-2Kb-

SIINFEKL (25-D1.16) (1:100). Stained cells were washed and then fixed with BD cytofix for 20 min at 4 °C, protected from light. In the case of intracellular staining, permeabilization was then performed by incubating with fixation-permeabilization buffer (ThermoFisher 88-8824-00) according to the manufacturer's recommendations prior to staining with intracellular targeted antibodies at the indicated dilutions in permeabilization buffer for 30 min at 4 °C and protected from light. Intracellular antibodies used: IL-2 (JES6-5H4) (1:100) and IFNγ (XMG1.2) (1:100). Cells were washed twice with permeabilization buffer and subsequently with cell staining buffer. For antigen-specific T-cell tetramer staining, either the Flex-TTM Biotin H-2 K(b) OVA Monomer (Biolegend 280051) paired with PE-streptavidin or APC-streptavidin (Biogened 405203 or 405207, respectively) was used according to manufacturers' instructions. Samples were acquired using a BD LSRII Fortessa. Downstream analysis was performed using TreeStar FlowJo, version 10.6.2. Representative flow cytometry gating strategies are detailed in the Supplementary Figs. t-distributed stochastic neighbor embedding was performed using FlowJo (version 10.8) with default settings to visualize high-dimensional flow cytometry data. Live tdTomato+ CD45+ cells from sentinel and non-sentinel lymph nodes were gated and downsampled equally prior to analysis.

### Tissue analysis

**Histology.** Tissue samples were fixed in zinc formalin fixative (Sigma-Aldrich) and sent to HistoServ, Inc. (Germantown, MD) for embedding, sectioning, and H&E staining. Histology samples were analyzed using QuPath 0.2.3, an open-source quantitative Pathology & Bioimage Analysis software (Edinburgh, UK). Immunohistochemistry on formalin-fixed paraffin-embedded lymph node samples or tumor samples was performed using anti-wide spectrum cytokeratin antibody (Abcam, ab9377, 1:200 dilution, overnight at 4 °C), CD8 (Abcam ab22378, 1:400 dilution overnight at 4 °C) or CD4 (ab183685, 1:400 dilution overnight at 4 °C). Tissues were then counterstained with biotinylated anti-rabbit secondary (Vector Labs, BA-1000, 1:400 dilution, 30 min at room temperature) or Goat Anti-Rat IgG H&L (HRP) (ab205720, 1:400, 30 min at room temperature). The protocol utilized is described in detail in ref. [59], with the following modifications (1) antigen retrieval was performed using low pH IHC Ag Retrieval Solution (ThermoFisher, 00-4955- 58) and subjected to heat using a steamer for 40 min, and (2) Bloxall Blocking Solution (Vector Labs, SP-6000, 20-min incubation, room temperature) was used to inactivate endogenous peroxidases. Slides were processed with either the ABC reagent (Vector Laboratories, # PK-6100) or the DAB substrate kit (Vector Laboratories, # SK-4105). Slides were scanned using a Zeiss Axioscan Z1 slide scanner equipped with a ×20/0.8 NA objective. All image analyses were performed using the QuPath software to perform pixel classification of stained cells.Multiplex immunofluorescenceSlides containing 4-µm sections were deparaffinized using a Leica autostainer (xylene−4 min; 100% ethanol−2 min; 95% ethanol−1 min; 70% ethanol−1 min; water). Antigen retrieval was performed in AR9 Buffer (AkoyaBioscience, AR900250ML) for 1 min (100% Power) and 10 min (10% Power) in a microwave. Endogenous peroxidase activity was eliminated using PeroxAbolish (BiocareMedical, PXA969) for 20 min. Slides were washed with distilled H₂O and TBS-T and blocked with goat serum (Vector Labs, S1000) for 20 min. Rabbit anti-CD11c (D1V9Y, Cell Signaling Technology, #97585, 1:250) was diluted in Renaissance antibody diluent (Biocare Medical, PD905L), added to the slide, and incubated for 45 min on an orbital shaker at RT. After washing in TBS-T, anti-rabbit secondary HRP (Vector Labs, MP-7451-15) was added for 15 min RT, and subsequently washed with TBS-T. Slides were incubated with Opal520 reagent at 1:150 dilution in 1× Plus Amplification buffer (Akoya Biosciences, NEL821001KT) for 10 min at RT and washed in TBS-T and distilled H2O. For the second cycle, slides underwent antibody stripping in Rodent Decloaker (Biocare Medical, RD913) for 1 min (100% Power) and 10 min

(10% Power) in a microwave, blocked with goat serum for 20 min RT, stained with rabbit anti-CD11b (Abcam, ab133357, 1:1000) for 45 min RT, anti-rabbit secondary HRP for 15 min RT, and Opal570 reagent (Akoya Biosciences, NEL821001KT) at 1:150 for 10 min RT. For the third cycle, slides underwent antibody stripping in Rodent Decloaker for 1 min (100% Power) and 10 min (10% Power) in a microwave, blocked with goat serum for 20 min RT, stained with rabbit anti-CD40 (E2Z7J, Cell Signaling Technology, #86165, 1:250) for 45 min RT, anti-rabbit secondary HRP for 15 min RT, and Opal620 reagent (Akoya Biosciences, NEL821001KT) at 1:250 for 10 min RT. For the fourth cycle, slides underwent antibody stripping in Rodent Decloaker for 1 min (100% Power) and 10 min (10% Power) in a microwave, blocked with goat serum for 20 min RT, stained with rabbit anti-CCR7 antibodies (Abcam, ab253187, 1:500) for 45 min RT, anti-rabbit secondary HRP for 15 min RT, and Opal690 reagent (Akoya Biosciences, NEL821001KT) at 1:100 for 10 min RT. After washes in TBS-T, DAPI (Akoya Biosciences, NEL821001KT) was added to slides for 10 min at RT. Slides were rinsed with TBS-T and distilled H2O and coverslipped with VectaShield Hard Mount (Vector Labs, H-1500-10). Slides were imaged at both ×10 and ×20 using the Vectra 3 Polaris (Akoya Biosciences). Acquired qpTIFF images from the Vectra Polaris system were imported into QuPath analysis software[60] and whole image analysis was performed using the object classification algorithm to obtain cell densities of targeted phenotypes.

### Chemokine array

Tumors and tumor-draining lymph nodes were isolated and tissue homogenate in lysis buffer (20 mM Tris HCl pH 7.5, 0.5% Tween 20, 150 mM NaCl) supplemented with protease inhibitors. Mouse Chemokine Array 44-Plex (MD44) was run by EVE Technologies (Calgary, AB, Canada).

### Murine sentinel lymph node mapping

Tilmanocept (Navidea Biopharmaceuticals, Dublin, OH) was conjugated to the near-infrared fluorescent dye IRDye800CW (LI-COR Biosciences, Lincoln, NE) following established protocols[61]. Purity of the resulting probe was confirmed using instant thin layer chromatography, with both radiochemical and fluorescent purities exceeding 98%. IRDye800CW exhibits an excitation maximum at 774 nm and emits at 789 nm, enabling high-sensitivity imaging with the Fluobeam800 system, comparable to the performance of ICG. For lymphatic mapping, 0.6 nmol (approximately 2 MBq in 10 μL volume) of either Tilmanocept-IRDye800CW or Lymphazurin Blue (Sigma, catalog #Sy3H98B979A9) was administered via submucosal injection at four peritumoral sites in the oral cavity (buccal mucosa or oral tongue). Animals were anesthetized or euthanized per IACUC-approved protocols at designated timepoints for SLN biopsy. Hair overlying the neck was shaved to improve visualization. For Lymphazurin, SLNs were visually identified by direct inspection under ambient light. For Tilmanocept-IRDye800CW, fluorescence imaging was performed using the Fluobeam800 camera with a 500 ms exposure time, which permitted detection of signal through intact soft tissue and prior to surgical exposure, as previously reported[62].

### In vivo mouse models and analysis

All the animal studies using HNSCC tumor xenografts and orthotropic implantation studies were approved by the University of California, San Diego (UCSD) Institutional Animal Care and Use Committee (IACUC), with protocol ASP #S16200, and all experiments adhere to all relevant ethical regulations for animal testing and research. All mice were obtained from Charles River Laboratories (Worcester, MA). Mice at UCSD Moores Cancer Center are housed in individually ventilated and micro-isolator cages supplied with acidified water and fed 5053 Irradiated Picolab Rodent Diet 20. The temperature for laboratory mice in this facility is mandated to be between 18 and 23 °C with

40–60% humidity. The vivarium is maintained in a 12-h light/dark cycle. All personnel were required to wear scrubs and/or lab coat, mask, hair net, dedicated shoes, and disposable gloves upon entering the animal rooms. WT C57Bl/6 mice were obtained from Charles River Laboratories (Worcester, MA). C57Bl/6 OT-1 (Tg-TcraTcrb-1100Mjb/J), BATF3 KO (Batf3tm1Kmm/J), R26-CreERT2 (ROSA26Sor^tm1(cre/ERT2)Tyj), and Ai9 (ROS)26Sor^tm9(CAG-tdTomato)Hze) animals were obtained from The Jackson Laboratory (Bar Harbor, ME).

**Tamoxifen treatment.** Tamoxifen was purchased from Sigma-Aldrich. Where indicated, mice were dosed with tamoxifen at 100 mg per kg (body weight) for systemic treatment or with 0.05 mg in 2.5 µL for intratumoral injection with a Hamilton GS microsyringe. A stock solution of 15 mg ml$^{-1}$ was prepared by dissolving tamoxifen in miglyol at 37 °C. After dissolving, the solution was stored at −20 °C protected from light. Analysis for tdTomato+ labeled cells occurred 48–120 hours after either systemic or local tamoxifen delivery in reporter animals.

**Local inflammation model.** To induce a localized inflammatory response, C57BL/6 mice were injected with 50 µg of ovalbumin (OVA, Sigma-Aldrich) and 10 mM 2′,3′-cGAMP (STING agonist, InvivoGen) in 50 µL of phosphate-buffered saline (PBS) into the buccal space. Control mice received PBS alone. Inflammation was allowed to develop over 72 hours before downstream experimentation.

**Orthotopic tumor modeling.** For orthotopic implantation, 4MOSC1 cells (1 × 10$^6$ per tumor) were injected into the oral cavity (tongue or buccal mucosa) of female C57Bl/6 mice (4–6 weeks of age). MOC1 cells (1 × 10$^6$ per tumor) were similarly implanted into the tongue. For drug treatment, mice received intraperitoneal or local injections, as specified for each experiment. Mice were monitored at least three times per week for tumor progression, body condition, grooming behavior, and weight loss. Tumor growth was assessed by caliper measurement. Experiments were terminated at predefined timepoints or earlier if mice met criteria for humane endpoints, including >20% weight loss, impaired mobility or grooming, visible distress, or tumor ulceration. According to ASP guidelines, euthanasia was also performed if tongue tumors exceeded 8 mm or buccal tumors exceeded 10 mm in greatest diameter, or in the event of ulceration. Tissues were subsequently collected for histological, immunohistochemical, flow cytometric, or sequencing analysis. The maximum tumor burden permitted by our institutional animal care protocol was not exceeded.

**In vivo depletion.** FTY720 (Cayman 10006292) 5 mg/kg/day IP was administered to inhibit lymphocyte egress from lymph nodes, and αCD8 antibodies (BioXCell, clone YTS169.4) 250 mg/mouse/dose IP were used for CD8 depletion every other day. MMP9 inhibition was achieved using a specific inhibitor (Sigma 444293) at 0.4 mg/mouse/dose delivered intratumorally in 8 µL volume on post-tumor transplant day 3, 5 and 7. SLN were mapped and harvested 48 hours after completion of treatment. This approach was designed to pre-condition the tumor microenvironment by limiting dendritic cell trafficking during the window of tdRT-induced immune activation, consistent with prior studies employing MMP9 inhibition to suppress immune priming via blockade of DC migration[38,40].

**Surgery.** All the animal surgery procedures were approved by the University of California, San Diego Institutional Animal Care and Use Committee (IACUC), with protocol #S16200. Mice were dosed with 0.1 mg/kg buprenorphine every 12 hours as needed for pain. Neck dissection and sentinel lymph node biopsy were performed: briefly, anesthetized animals were positioned and draped and prepped in sterile fashion. Under ×8 operative microscopy, the skin was incised sharply in the midline with straight microscissors and skin flaps were bluntly elevated laterally to broadly expose the cervical space spanning from the angle of the mandible bilaterally to the clavicles. Superficial lymphatic basins or the SLN after mapping were encountered immediately deep to the dermis and adjacent to the super-olateral aspects of the submandibular glands and were liberated with blunt dissection and handheld monopolar cautery from surrounding tissues. Reflecting the submandibular glands and superficial lymphatic basins laterally revealed the jugular venus plexus and deep lymphatic basins nested within the jugular vascular confluence and atop the floor of the neck. Deep lymphatic tissues were resected after blunt dissection to liberate them from surrounding tissues. After resection, hemostasis was confirmed or achieved with cautery. For LCA, previously mapped animals were anesthetized and skin incised. Mapped channels were visualized under microscopy and ligated sharply with microscissors. Native tissues were repositioned, and the wound was closed in a single layer with 5-0 simple interrupted vicryl sutures. Animals were placed under a heating lamp in a recovery space and observed until fully conscious. For the sham surgery group, mice were anesthetized, and skin flaps were raised with care to not disturb underlying lymphatic channels; no tissues were resected in the sham group animals.

**Radiation.** The dedicated small animal radiotherapy planning system SmART-Plan (version 1.3.1, Precision X-ray, North Branford, CT) was used to create, evaluate, and deliver irradiation. Animals were anesthetized with isoflourane and positioned within the SmART machine, secured to the stage. A spiral CT scan with 1mm cuts of the neck was obtained, and cervical lymphatics were delineated as the planning target volume. A 5 mm collimator was installed, and two static parallel opposed beams linked to the irradiator isocenter were used to deliver homogenous single fraction doses to the planned target volume. Radiation doses were as indicated in the figure legends (4, 8, or 12 Gy to tumor and 4 Gy to draining lymph nodes for elective nodal irradiation [ENI]).

## RNA sequencing and analysis

**RNA isolation.** Tissues were harvested as described above. Samples were transferred into microcentrifuge tubes containing 1 mL TRIzol. RNA was then isolated using Qiagen RNeasy Mini Columns (74004; Qiagen, Germantown, MD) according to the manufacturer's recommendations and including an on-column DNase I digestion. Yield and integrity of RNA were confirmed by reading absorbance at 260, 280, and 230 nm using a NanoDrop ND-1000 (NanoDrop Technologies; ThermoFisher Scientific, Inc., Wilmington, DE, USA) and with the Agilent 2200 Tapestation (Agilent Technologies, Inc.). Library preparation and paired-end 150 bp (PE150, Illumina) RNA sequencing were performed by Novogene (Novogene Corporation, Sacramento, USA).

**Cite-sequencing.** CITE-sequencing was performed using the BioLegend TotalSeq™ C Universal Kit (BioLegend, San Diego, CA) to simultaneously profile cell surface proteins and transcriptomes. Sentinel lymph nodes were prepared according to the manufacturer's protocol, and the 10x Genomics Chromium platform was utilized to generate single-cell libraries. Sequencing was conducted at the La Jolla Institute for Immunology's Sequencing Core Facility using the NovaSeq 6000 (S10OD025052) system, and sequencing data were processed and analyzed as described below.

**TCR-sequencing.** Genomic DNA was extracted from the tumor and sentinel lymph nodes using the Qiagen DNeasy Blood & Tissue Kit (Qiagen, Hilden, Germany) according to the manufacturer's instructions. The extracted gDNA was then sent to Adaptive Biotechnologies (Seattle, WA) for TCR sequencing, following their established protocols. Prior to submission, we conducted quality control assessments to ensure the integrity and concentration of the gDNA, with all samples

exceeding minimum thresholds recommended by Adaptive Biotechnologies. Sequencing. Data generated was analyzed using their proprietary software, hosted on the ImmunoSeq Analyzer Website (https://clients.adaptivebiotech.com/).Alignment/differential expressionPaired-end reads were aligned using STAR v2.7.9 using default options. STAR index was created using the GRCm39 primary genome FASTA and annotation files. The resulting BAM files were sorted by name using samtools v1.7 then gene counts were quantified using HTSeq-count v0.13.5. Pairwise differential expression was calculated and PCA plots were created using DESeq2 v1.32.0.

**GSEA/GO**. Gene set enrichment analysis was conducted using the GSEAPreranked v7.2.4 module on the GenePattern public server, gsea-msigdb.org, with 10,000 permutations, and the genes mapped and collapsed to standard mouse symbols using the MSigDB mapping file version v7.4[63]. The Gene Ontology (Biological Processes) and ImmunesigDB gene set collections were used[64]. The ranked list of genes was created using the log2-fold change (FC) calculated by DESeq2. For this analysis, genes more highly expressed in late relative to early neck dissection are at the top of the ranked list. Gene ontology (GO) analysis was performed through the GeneOntology.org website using the top significant (log2FC > 1, P value < 0.05) upregulated genes in the samples from the late neck dissection group.

**Single-cell RNAseq analysis**. Quality control, alignment, and quantification of reads were performed using Cell Ranger v.(7.0.1) software from 10X Genomics. Mouse sequences were aligned to the mouse reference genome prepared by 10X Genomics (mm10-2020-A GENCODE vM23/Ensembl 98). Downstream analysis was performed with Seurat v4 (PMID: 34062119). Quality filtering was performed on the cells, with cells being retained if they contained more than 600 features, fewer than 4000 features, and less than 10% mitochondrial reads. Reads were normalized usin SCTransform (v2) (PMID: 31870423). Samples were integrated using the FindIntegrationAnchors procedure implemented in Seurat V4. Cells were clustered using the Louvain modularity optimization algorithm. Cell types were assigned based on automated cell type annotation tools (SingleR (PMID: 30643263) and CellTypist (PMID: 38134877)), along with being informed by prior biological knowledge of expected cell types and marker genes. Cell type marker genes were computed using the FindAllMarkers function in Seurat. Subtyping of cell types was performed using Seurat's FindSubCluster function.

**BioRender statement**
Some figures were created with BioRender.com. Publication licenses have been obtained in accordance with BioRender's academic licensing agreement. Appropriate attribution has been included in each relevant figure legend.

**Statistics and reproducibility**
Data analysis was performed with GraphPad Prism version 9 for Mac. The differences between experimental groups were analyzed using independent t tests, one- or two-way ANOVA with multiple comparisons, Fisher's exact test, DESeq2, Log2FC $P < 0.05$, or simple linear regression analysis as indicated. Survival analysis was performed using the Kaplan–Meier method and log-rank tests. The asterisks in each figure denote statistical significance, or ns for non-significant ****$P < 0.0001$. All the data are reported as mean ± SEM (standard error of the mean). For all experiments, each experiment was independently repeated at least twice with similar results.

**Reporting summary**
Further information on research design is available in the Nature Portfolio Reporting Summary linked to this article.

## Data availability
Source data are provided with this paper. The bulk RNA sequencing data generated in this study have been deposited in the Sequence Read Archive (SRA) under BioProject accession code PRJNA1183332. The CITE sequencing and TCR sequencing datasets generated in this study are available in the Gene Expression Omnibus under accession codes GSE276437 and GSE276434, respectively. Source data are provided with this paper.

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

## Acknowledgements

We thank Dr. Ezra Cohen, Professor of Medicine and Associate Director of Translational Science at UC San Diego Moores Cancer Center; Dr. Jack Bui, Professor of Pathology at UC San Diego; Dr. John Chang, Professor of Medicine in the Division of Gastroenterology at UC San Diego; and Dr. Ravi Uppaluri, Chief of Otolaryngology–Head and Neck Surgery at Brigham and Women's Hospital and Professor at Harvard Medical School, for their insightful discussions and guidance throughout the development of this study. Their mentorship and feedback were invaluable in shaping the experimental design and interpretation of results.

## Author contributions

R.S.-K. and J.A.C. conceived and supervised the study. R.S.-K., R.A.M., S.T., C.P., S.S., L.M.C., S.L., S.F., R.J., and I.F.P. performed experiments and data collection. F.F., S.S., B.S.Y., M.M.A., and K.E.D. contributed to experimental design and data interpretation. C.A.N., D.C.-F., and S.B.R. assisted with computational and single-cell data analysis. S.M.J., B.A.F., and R.B.B. provided conceptual guidance and contributed to study design and manuscript editing. J.S.G., A.S., and J.A.C. contributed to data interpretation and manuscript revision. R.S.-K. wrote the manuscript with input from all authors. All authors reviewed and approved the final manuscript.

## Competing interests

The authors declare no competing interests.

## Additional information

Robert Saddawi-Konefka ⓘ[1,2,3] ✉, Riyam Al Msari[2,3], Shiqi Tang[2,3], Chad Philips[2,3], Sayed Sadat[2,3], Lauren M. Clubb[2,4], Sarah Luna ⓘ[2,3], Santiago Fassardi[2,3], Riley Jones ⓘ[2,3,5], Ida Franiak Pietryga[2,3,5], Farhoud Faraji ⓘ[1,2,3], Shiruyeh Schokrpur ⓘ[2,3,6], Bryan S. Yung[2,4], Michael M. Allevato[2,4], Kelsey E. Decker[2,3], Chanond A. Nasamran[7], Daisy Chilin-Fuentes[7], Sara Brin Rosenthal ⓘ[7], Shawn M. Jensen ⓘ[8,9], Bernard A. Fox ⓘ[8,9], R. Bryan Bell[8,9], J. Silvio Gutkind[2,3,4], Andrew Sharabi[2,3,5] & Joseph A. Califano ⓘ[1,2,3] ✉

[1]Department of Otolaryngology-Head and Neck Surgery, UC San Diego School of Medicine, San Diego, CA, USA. [2]Moores Cancer Center, UC San Diego, La Jolla, CA, USA. [3]Gleiberman Head and Neck Cancer Center, Moores Cancer Center, UC San Diego, La Jolla, CA, USA. [4]Department of Pharmacology, UC San Diego, La Jolla, CA, USA. [5]Department of Radiation Medicine and Applied Sciences, UC San Diego School of Medicine, San Diego, CA, USA. [6]Department of Medicine, Division of Hematology-Oncology, UC Davis, Sacramento, CA, USA. [7]Center for Computational Biology & Bioinformatics, Department of Medicine, University of California, San Diego, La Jolla, CA, USA. [8]Earle A Chiles Research Institute, Robert W Franz Cancer Research Center, Providence Portland Medical Center, Portland, OR, USA. [9]Department of Molecular Microbiology and Immunology, Oregon Health Science University, Portland, OR, USA. ✉e-mail: rsaddawi@health.ucsd.edu; jcalifano@health.ucsd.edu

