## [Transparent Peer Review file · Nature Communications]

The Tumor-Sentinel Lymph Node Immune Migratome Reveals a Key Role for Migratory CCR7+ Dendritic Cells in Response to Sequenced Immunoradiotherapy

Corresponding Author: Professor Joseph Califano

Version 0:

Reviewer comments:

Reviewer #1

(Remarks to the Author)

The authors report on a study examining tdRT (tumor directed radiotherapy) plus anti-PD1 therapy on tumor lymph nodes (LNs) and dendritic cell dynamics in head and neck cancer models. Their previous important work showed LN and specifically cDC1s in these structures are critical for checkpoint responses in head and neck cancer models. Here, they focus on tdRT and understanding role of immune cell migration between tumor and lymph nodes with radiation plus checkpoint blockade therapy. The work extends from detailed mapping of sentinel lymph nodes, a labeling approach to identify intratumoral immune cells, and characterization of immune responses with tdRT+ antiPD1 therapy.

My main concerns are that these data are incremental to their previous study and the implication that sequenced tdRT enhances DC mediated priming to improve antiPD1 therapy is not supported. They already established the role of SLNs and DCs in the LNs for ICI response so it is expected that adding tdRT to ICI in these same models would have the same SLN/DC dependency. The statement that tdRT shifts DC from "their canonical role in promoting tolerance to driving antitumor immunity" is not supported as a tolerance role is not shown at baseline and a direct functional impact is also not shown. Overall, authors are suggesting that sequenced tdRT, via DC migration, primes a CD8+ T cell response that then enhances anti-PD1 therapy. No data are shown that not sequencing tdRT (give anti-PD1 before tdRT for example) is not efficacious. It is not clear to me that tdRT given on day 6 with antiPD1 on day 8/10 has the temporal characteristics of a primed response (on average would be ~5-7 days). The data in Fig 4F show tumors already rejecting at day 8/9.

1. Introduction cites Ma et al study of tdRT+antiPDL1/CTLA4 eliminating elective nodal radiation in HPV+ HNSCC. Please note, this study showed a 26% locoregional recurrence rate at a median of 3 months suggesting great care should be taken with this approach of not radiating nodal basin- please acknowledge this including in the Discussion along with discussion of other studies discussed.
2. Overall, Figure 1A/B is a very detailed mapping study. Mapping of SLN with 2 agents- is there a difference in buccal versus tongue drainage patterns? These data could be moved to supplementary as I don't think contribute to overall story as presented.
3. 1D aims to show that tamoxifen does not migrate to SLN within 2 hours after tongue injection. I am unclear that this experiment shows that tamoxifen does not drain to LNs and activate Cre in LNs. Tamoxifen may not drain in 2 hours, but could tamoxifen migrate to SLN and induce tdTomato 72 hrs later (as for experimental approach in 1H or 5E)? This is a central premise that authors have identified a tumor migratome but it is possible that some tamoxifen drains regionally with longer exposure. Other approaches have been described that could avoid this confounding issue (Kaede mice or labeling the tumor with ZsGreen)
4. 1E- what is the role of OVA injection as tdTomato was assessed by FACS?
5. Fig 1H shows a diverse migratome. Have authors tested (by flow cytometry) whether there is a different or similar proportion/composition in CD45+tdTomato+ cells in nSLNs? Fig 1F shows that the nSLNs have about 50% fewer tdTomato cells compared to SLNs and the statement that the SLN harbors "a unique immunologic niche" is not fully supported.
6. Fig 2 shows that a single sub-cytotoxic dose (4Gy) of tumor directed RT results in altering the tumor microenvironment. These data are confirmatory of other studies showing high dose radiation can modify the TME (authors cite examples, but should also include PMID: 29399393). The statement that tdRT changes "the canonical, pro-tolerogenic activity of migratory dendritic cells towards an antitumor role" is not fully supported. There is a modest increase in CD8s expressing CXCR3 and

similarly in CCR7 expressing DCs. For the latter, there is a change from average of ~10% to ~12% - this is statistically significant but am not sure it is functionally meaningful. No FACS comparison is shown to a wider field radiation so saying RT that is "lymphatic sparing" is specifically driving these changes cannot be stated. Note that in their previous work, 18Gy radiation to the LN abrogated ICI response. Does the same 4 Gy dose at a wider field induce same changes as tdRT. The H-2kb-SIINFEKL antibody is known to have high background staining. No gating for this is shown in supplementary data. I think a more informative experiment may be to examine for enhanced SIINFEKL responses in draining nodes with the tdRT and the LucOS system.

7. Fig 3 focuses on changes with tdRT and shows a rationale for combining with antiPD1. The RNASeq pathway analysis compares untreated to tdRT+antiPD1. Are these significantly different than pathways in 2E? What is the impact of antiPD1 alone on these pathways? There is a clear response to combination tdRT/ antiPD1 in both the 4MOSC1 and 4 MOSC2 models but antiPD1 therapy by itself results in 44% response in 4MOSC1 and a nearly 30% response in 4MOSC2. The statement that "sub-cytotoxic, tumor-directed radiation prior to immunotherapy upregulates programs of antitumor immune surveillance and dramatically enhance the efficacy of α PD-1 ICI therapy" is not fully supported. The statement would be supported if data are shown that tdRT concurrent with anti-PD1 or anti-PD1 followed by tdRT did not improve response.

8. Figure 4- these data confirm that LNs are important for response, and to me represent an extension of previous work but here with tdRT + anti-PD1. Please clarify dosing used here- was 4Gy dosing used for LN radiation as well as tdRT?

9. Fig 5- SLN studies are then shown where again, changes in 5B should include anti-PD1 therapy alone comparisons. There are modest changes in CCL19 levels and DC levels (latter should be shown as % of CD11c in 5D as in Figure 2F- unclear if these are significant to have a functional impact). As stated above, I am not clear that these Fig 5E data truly represent a "tumor migratome" and also there is no comparison to anti-PD1 alone. Authors again state that DCs have been functionally reprogrammed. An experiment to show functional confirmation and supporting this statement could involve harvesting DCs from SLN of mice bearing LucOS expressing tumors, performing tdRT +/-antiPD1 and assessing ex vivo OT1 T cell priming.

10. Experiments in 6 further show importance of DCs. 6B DC gating/ percentages should be consistent with Figures 2F and 5D. Please clarify what is the "Normalized" percentage in 6C, are the percentages in the representative plot of the total cells or of TCRbeta? 6E shows experiments with an MMP9 inhibitor. I am not familiar with this approach but the inhibitor is delivered to the primary tumor on days 3, 5 and 7 with tdRT on day 6 (as per other figures). This seems counterintuitive with tdRT inducing DC migration on day 6 and beyond but DC migration is already being inhibited 3 days prior to therapy that is meant to activate them.

Minor point

1. Supp Fig 2C legend states "Tumor growth curves for control, targeted field, and large field tdRT-treated groups (4Gy, 8Gy, 12Gy tdRT)." All figures are labeled tdRT but large field would encompass LNs- correct?
2. Supp Fig 2D- is labeling of axes incorrect for the Texh and Tpex subsets?

Reviewer #2

(Remarks to the Author)

The work presented demonstrates the role of the sentinel lymph node in priming of anti-tumor T cell responses in orthotopic preclinical models of HNSCC treated with focal radiotherapy. The pro-immunogenic effect of radiation is achieved with a single 4 Gy dose, which by itself has no effects on the tumor growth but can significantly increase the response to anti-PD1. This work is a follow up to prior work by the authors and by others highlighting the importance of sparing draining nodes (and especially the sentinel node) when radiation therapy is used with the intent of increasing responses to immunotherapy. In this manuscript new mechanistic insights are provided about the role of the draining node in the pro-immunogenic effects of radiation therapy.

The experiments are sound and elegant and leverage genetic and pharmacologic tools to present a detailed analysis of the role of dendritic cells that migrate between the tumor and the lymph node in the response to combinations of tumor-targeted radiation and anti-PD1.

However, the translational significance and scientific novelty could be further enhanced in some areas, as detailed below.

1) As recognized by the authors in Discussion, the use of a single 4 Gy dose is a limitation as this is not clinically relevant. The two clinical trials that inspired – in part – this work (refs. 7 and 8) use doses of 6 to 8 Gy repeated 2 to 5 times in combination with anti-PD1. Thus, it seems a missed opportunity not to investigate if using other fractionated regimens used in the clinic (6 Gy x 2 or 6 Gy x 3 or 8 Gy x 2 and 8 Gy x 3) could increase cure rates with anti-PD1 above what is observed with 4Gy single dose, or rather, show potentially detrimental effects. One question that is extremely interesting and could be addressed using the R26-CreERT2 x Ai9 and local injection of tamoxifen is what is the optimal spacing between radiation doses (24 hours, 48 hours?) to enable the generation and migration of CCR7+ DC to the SLN.

2) Figure 1H and Supplemental Fig 1F show that the SLN immune migratome includes, besides DC, distinct populations of CD4+ T cells, CD8+ T cells, B cells, and NK cells. However, this important observation is not further developed. It seems important to know what the B cell clusters are – whether they could perform regulatory function – as previously described (PMID 37344597) or may serve as antigen-presenting cells. Similarly, the nature of the T cells that are labelled by tDTomato and move from the tumor to the SLN is of interest. What happens to these populations when the tumor is irradiated and/or treated with anti-PD1?

3) Figure 4B: is surgery immunosuppressive, at least temporarily? What happens if surgery is performed but the SLN not removed?

4) A minor issue: Supplementary Figure 5E mentioned on page 9 does not exist.

Version 1:

Reviewer comments:

Reviewer #1

(Remarks to the Author)

Thank you to the authors for carefully addressing my critiques. With the exception of some needed clarification on a couple of my comments, authors have addressed all of my other critiques.

For Comment 3, I wanted clarification on whether injecting tamoxifen and letting it drain for 72 hours would result in labeling of tdTomato cells in the SLN. The experiment as presented shows that ovalbumin/STING are injected and 72 hours later tamoxifen is injected. How long after this 72 hr injection were the SLN analyzed with and without lymphatic channel ablation? Was this done 48-72 hours after tamoxifen as stated in methods line 698? Please include this information in the schema/ legend.

For Comment 6, I had requested gating controls for the H-2Kb-SIINFEKL antibody (25-D1.16) used in Figure 2H. I was not referring to the H2Kb-SIINFEKL tetramer specificity (as shown in Fig S2J). I think Lines 218-219 stating that the OT-1 T cell gating strategy in Fig S2J which shows tetramer specificity does not apply to the Figure 2H.

Reviewer #2

(Remarks to the Author)

The authors have satisfactorily addressed my prior questions and the manuscript is improved and overall provides important new information.

Point-by-Point Response

Reviewer #1

General Comment: “The authors report on a study examining tdRT (tumor directed radiotherapy) plus anti-PD1 therapy on tumor lymph nodes (LNs) and dendritic cell dynamics in head and neck cancer models. Their previous important work showed LN and specifically cDC1s in these structures are critical for checkpoint responses in head and neck cancer models. Here, they focus on tdRT and understanding role of immune cell migration between tumor and lymph nodes with radiation plus checkpoint blockade therapy. The work extends from detailed mapping of sentinel lymph nodes, a labeling approach to identify intratumoral immune cells, and characterization of immune responses with tdRT+ antiPD1 therapy.

My main concerns are that these data are incremental to their previous study and the implication that sequenced tdRT enhances DC mediated priming to improve antiPD1 therapy is not supported. They already established the role of SLNs and DCs in the LNs for ICI response so it is expected that adding tdRT to ICI in these same models would have the same SLN/DC dependency. The statement that tdRT shifts DC from “their canonical role in promoting tolerance to driving antitumor immunity” is not supported as a tolerance role is not shown at baseline and a direct functional impact is also not shown. Overall, authors are suggesting that sequenced tdRT, via DC migration, primes a CD8+ T cell response that then enhances anti-PD1 therapy. No data are shown that not sequencing tdRT (give anti-PD1 before tdRT for example) is not efficacious. It is not clear to me that tdRT given on day 6 with antiPD1 on day 8/10 has the temporal characteristics of a primed response (on average would be ~5-7 days). The data in Fig 4F show tumors already rejecting at day 8/9.”

We thank the reviewer for these thoughtful comments. In response, we performed new experimental comparisons and analyses. Most saliently, as presented in Figure 4A–D, we directly assess the impact of treatment sequencing. We find that tdRT followed by α PD-1 leads to 9/10 complete responses, compared to 3/10 with reverse sequencing. Moreover, sequenced therapy is associated with significantly increased infiltration of antigen-specific T cells in the tumor and the presence of tumor-specific antigen-presenting cells in the SLN. Together, these results underscore the mechanistic importance of treatment order. We have revised the text accordingly to reflect these findings.

Comment 1: “Introduction cites Ma et al study of tdRT+antiPDL1/CTLA4 eliminating elective nodal radiation in HPV+ HNSCC. Please note, this study showed a 26% locoregional recurrence rate at a median of 3 months suggesting great care should be taken with this approach of not radiating nodal basin- please acknowledge this including in the Discussion along with discussion of other studies discussed.”

We thank the reviewer for this helpful comment. We have revised the text to clarify that, while Ma et al. and other early-phase trials suggest that sparing uninvolved lymphatics can enhance responses to immunoradiotherapy, the 26% locoregional recurrence rate at early follow-up underscores the need for caution. This finding reinforces the importance of better understanding the mechanisms of durable immunity and optimizing the timing and extent of nodal irradiation. We have incorporated this point into the Discussion alongside related studies.

Comment 2: “Overall, Figure 1A/B is a very detailed mapping study. Mapping of SLN with 2 agents— is there a difference in buccal versus tongue drainage patterns? These data could be moved to supplementary as I don’t think contribute to overall story as presented.”

We thank the reviewer for this helpful suggestion. In response, we have moved the SLN mapping data—including dual-tracer visualization and drainage quantification—to the supplemental figures (Supplemental Fig. 1A–E). These panels establish the anatomical basis and reproducibility of SLN identification, which support all downstream mechanistic studies. Importantly, we observed no difference in the drainage patterns between buccal and tongue tumor injections, indicating consistent SLN targeting across oral cavity subsites. We agree that this allows the main figures to focus more directly on the core immunologic findings.

Comment 3: “1D aims to show that tamoxifen does not migrate to SLN within 2 hours after tongue injection. I am unclear that this experiment shows that tamoxifen does not drain to LNs and activate Cre in LNs. Tamoxifen may not drain in 2 hours, but could tamoxifen migrate to SLN and induce tdTomato 72 hrs later (as for experimental approach in 1H or 5E)? This is a central premise that authors have identified a tumor migratome but it is possible that some tamoxifen drains regionally with longer exposure. Other approaches have been described that could avoid this confounding issue (Kaede mice or labeling the tumor with ZsGreen).”

We thank the reviewer for this thoughtful point. We agree that it is essential to rigorously establish the specificity of tdTomato labeling in the tumor, as this underpins our central premise of defining a tumor→SLN migratome. To test whether tamoxifen might drain to the SLN and label cells outside of the tumor over a longer period of time, we employed a previously described model of localized oral cavity inflammation using ovalbumin inoculation. In this setting, surgical interruption of the afferent lymphatic channel 72 hours after inoculation prevented tdTomato+ CD8+ T cell labeling in the SLN (Main Fig. 1C), confirming a unidirectional and requisite migration from the periphery to the SLN. These findings demonstrate that tdTomato+ cells present in the SLN does not result from delayed passive diffusion of tamoxifen in this model and help validate the use of tdTomato as a faithful marker of migratory immune cells in our system.

Comment 4: “1E- what is the role of OVA injection as tdTomato was assessed by FACS?”

We appreciate the reviewer’s question. Ovalbumin was used in this experiment as a surrogate antigen to induce localized immune activation in the oral cavity, consistent with our prior and other publications. This allowed us to study immune cell migration under conditions of antigen-driven inflammation without confounding the labeling of tdTomato+ cells. This rationale has been clarified with additional references added in the revised text.

Comment 5: “Fig 1H shows a diverse migratome. Have authors tested (by flow cytometry) whether there is a different or similar proportion/composition in CD45+tdTomato+ cells in nSLNs? Fig 1F shows that the nSLNs have about 50% fewer tdTomato cells compared to SLNs and the statement that the SLN harbors 'a unique immunologic niche' is not fully supported.”

We thank the reviewer for this important point. As shown in Fig. 1D (previous Fig 1F that the reviewer notes in the comment), we previously observed a significantly greater proportion of live tdTomato+ cells in the SLN compared to the nSLN and spleen, consistent with selective trafficking through afferent lymphatics to the SLN. To directly address the reviewer’s question regarding the composition of these tdTomato+ migratory cells, we performed new flow cytometry experiments comparing the immune subsets among tdTomato+ CD45+ cells in SLNs versus nSLNs or axillary lymph node. These new data, now shown in Fig. 1E–F and Supplemental Fig. 1F, reveal that the SLN harbors a unique population of migratory immune cells, including MHCII^{hi} activated dendritic cells and progenitor-exhausted CD8+ T cells (Tpex), that are not observed at comparable frequency in nSLNs or axillary lymph nodes. While T cells are present in higher abundance in nSLNs, they are largely naïve-like and do not exhibit the same functional phenotypes as tdTomato+ cells in the SLN (Supplemental Fig 1F). These findings support the conclusion that the SLN hosts a distinct immunologic niche shaped by selective immune trafficking from the tumor.

Comment 6: “Fig 2 shows that a single sub-cytotoxic dose (4Gy) of tumor directed RT results in altering the tumor microenvironment. These data are confirmatory of other studies showing high dose radiation can modify

the TME (authors cite examples, but should also include PMID: 29399393). The statement that tdRT changes 'the canonical, pro-tolerogenic activity of migratory dendritic cells towards an antitumor role' is not fully supported. There is a modest increase in CD8s expressing CXCR3 and similarly in CCR7 expressing DCs. For the latter, there is a change from average of ~10% to ~12%- this is statistically significant but am not sure it is functionally meaningful. No FACS comparison is shown to a wider field radiation so saying RT that is 'lymphatic sparing' is specifically driving these changes cannot be stated. Note that in their previous work, 18Gy radiation to the LN abrogated ICI response. Does the same 4 Gy dose at a wider field induce same changes as tdRT. The H-2Kb-SIINFEKL antibody is known to have high background staining. No gating for this is shown in supplementary data. I think a more informative experiment may be to examine for enhanced SIINFEKL responses in draining nodes with the tdRT and the LucOS system.”

We thank the reviewer for these thoughtful comments, as addressing them has strengthened the impact of our findings. We have refined our interpretation of dendritic cell function to reflect that the observed shifts in CCR7+ MHCII+ populations—while modest—are consistent with early activation programs and align with the broader immunomodulatory effects of tdRT and updated the text accordingly. To evaluate the role of lymphatic sparing, we reference data (Fig. 4E–F) comparing wide-field radiation to tumor-directed radiation (tdRT), both delivered at 4 Gy, demonstrating that tdRT more effectively potentiates antitumor response, consistent with a functional consequence of lymphatic sparing. In addition, we also confirm the functional relevance of tdRT-induced antigen presentation through enhanced SIINFEKL-specific responses in draining lymph nodes using the LucOS model (Fig. 2H), and we have now included appropriate gating controls for H-2Kb-SIINFEKL (FMO, wild-type, and OT-1) in Supplemental Fig. 2J. Finally, the suggested reference (PMID: 29399393) has been added to the revised text.

Comment 7: “Fig 3 focuses on changes with tdRT and shows a rationale for combining with antiPD1. The RNASeq pathway analysis compares untreated to tdRT+aPD1. Are these significantly different than pathways in 2E? What is the impact of antiPD1 alone on these pathways? There is a clear response to combination tdRT/antiPD1 in both the 4MOSC1 and 4 MOSC2 models but antiPD1 therapy by itself results in 44% response in 4MOSC1 and a nearly 30% response in 4MOSC2. The statement that 'sub-cytotoxic, tumor-directed radiation prior to immunotherapy upregulates programs of antitumor immune surveillance and dramatically enhance the efficacy of α PD-1 ICI therapy' is not fully supported. The statement would be supported if data are shown that tdRT concurrent with anti-PD1 or anti-PD1 followed by tdRT did not improve response.”

We thank the reviewer for these thoughtful comments, which helped elevate the mechanistic insights and translational significance of our study. To clarify the distinct contributions of α PD-1 and tdRT to the antitumor immune response, we performed RNA-seq analysis across all treatment conditions. α PD-1 alone had minimal transcriptional impact (Supplemental Fig. 3B), while tdRT monotherapy engaged a subset of immune programs (Figure 2E). In contrast, tdRT followed by α PD-1 induced robust activation of antigen processing and presentation, phagocytosis, and T cell activation pathways (Figure 3C, Supplemental Fig. 3C). Additionally, comparison of sequenced therapy (tdRT \rightarrow α PD-1) with tdRT alone revealed expanded enrichment of lymphocyte activation pathways (Supplemental Fig. 3D). These data support that tdRT primes the host, enabling enhanced responsiveness to checkpoint inhibition.

To evaluate whether treatment order influences the therapeutic benefit of combination immunoradiotherapy, we compared outcomes in animals receiving α PD-1 prior to tdRT versus tdRT followed by α PD-1 (Figure 4A). Tumor growth analysis revealed that sequencing tdRT prior to α PD-1 resulted in a significant improvement in tumor control compared to α PD-1 \rightarrow tdRT or monotherapy (Figure 4B; Supplemental Figure 4A). To better understand how this treatment order enhances efficacy, we examined antigen-specific immune responses. We observed a marked increase in OVA-specific CD8+ T cells in tumors of animals treated with tdRT \rightarrow α PD-1 compared to the reverse order, as assessed by tetramer staining (Figure 4C). In parallel, we detected an increased frequency of H-2Kb-SIINFEKL+ cross-presenting APCs in the sentinel lymph node (Figure 4D), suggesting enhanced priming of antigen-specific T cells. These findings demonstrate that sequencing tdRT prior

to α PD-1 therapy upregulates antigen presentation and primes host antitumor immune responses, thereby enhancing the efficacy of immune checkpoint blockade.

Comment 8: “Figure 4—these data confirm that LNs are important for response, and to me represent an extension of previous work but here with tdRT + anti-PD1. Please clarify dosing used here—was 4Gy dosing used for LN radiation as well as tdRT?”

We thank the reviewer for this clarification request. All radiation groups shown in Figure 4—including tumor-directed and elective nodal irradiation—received 4 Gy. We have updated the figure legend and Methods to clarify this

Comment 9: “Fig 5—SLN studies are then shown where again, changes in 5B should include anti-PD1 therapy alone comparisons. There are modest changes in CCL19 levels and DC levels (latter should be shown as % of CD11c in 5D as in Figure 2F—unclear if these are significant to have a functional impact). As stated above, I am not clear that these Fig 5E data truly represent a 'tumor migratome' and also there is no comparison to anti-PD1 alone. Authors again state that DCs have been functionally reprogrammed. An experiment to show functional confirmation and supporting this statement could involve harvesting DCs from SLN of mice bearing LucOS expressing tumors, performing tdRT +/-antiPD1 and assessing ex vivo OT1 T cell priming.”

We thank the reviewer for raising this important point, as we agree that direct evidence of functional reprogramming is essential to support our claim. To rigorously assess dendritic cell reprogramming, we examined T cell priming and trafficking to the tumor in vivo using the 4MOSC1-LucOS model. We chose this particular model as it afforded an opportunity to simultaneously track and quantify H-2Kb-SIINFEKL+ APCs and OT1-Tetramer+ CD8+ T cells, which collectively represent in vivo functional capacity of migratory dendritic cells involved in the host antitumor response. This approach, shown in Fig. 4C–D, revealed a higher frequency of H-2Kb-SIINFEKL+ APCs in the SLN and OT1-Tetramer+ CD8+ T cells infiltrating into tumors following tdRT \rightarrow α PD-1 treatment. This observation aligns with our prior findings in Supplemental Fig. 5E, which demonstrate increased H-2Kb-SIINFEKL+ MHCII+ dendritic cells in the sentinel lymph node following combination therapy. Together, these data support in vivo functional evidence for dendritic cell reprogramming following sequenced tdRT \rightarrow α PD-1.

To further clarify the composition of the tumor immune migratome, we refer the reviewer to the new data presented in Main Fig. 1E-F and Supplemental Fig. 1I, which demonstrates that tdTomato+ migratory immune cells localize preferentially to the SLN and include a distinct population of MHCII^{hi} dendritic cells and T_{pex} CD8+ T cells. To evaluate how therapy modifies this migratory signature, we performed CITE-seq analysis of tdTomato+ CD8+ T cells in the SLN following tdRT \rightarrow α PD-1. As shown in Supplemental Fig. 5H, this analysis revealed increased expression of genes associated with activation, cytokine signaling, and interferon responses, suggesting that therapy modulates the phenotype and function of migratory T cells arriving in the SLN.

We also thank the reviewer for highlighting the importance of directly comparing transcriptional responses to α PD-1 monotherapy. In Supplemental Fig. 5A–D, we now show that α PD-1 alone has minimal effect on immune activation or dendritic cell function in the SLN, in contrast to tdRT \rightarrow α PD-1, which significantly enhances pathways related to antigen presentation, phagocytosis, and DC maturation (Fig. 5A). Lastly, flow cytometry data in Figures 5C have been updated to show dendritic cell activation as a percentage of CD11c+ cells, as requested.

Comment 10: “Experiments in 6 further show importance of DCs. 6B DC gating/percentages should be consistent with Figures 2F and 5D. Please clarify what is the 'Normalized' percentage in 6C, are the percentages in the representative plot of the total cells or of TCRbeta? 6E shows experiments with an MMP9 inhibitor. I am not familiar with this approach but the inhibitor is delivered to the primary tumor on days 3, 5 and 7 with tdRT on day 6 (as per other figures). This seems counterintuitive with tdRT inducing DC migration on day 6 and beyond but DC migration is already being inhibited 3 days prior to therapy that is meant to activate them.”

We thank the reviewer for this detailed and thoughtful comment. We have revised Main Fig. 6B to report DC gating as a percentage of CD11c⁺ cells, consistent with Figures 2F and 5D. In Figure 6C, the “normalized” percentages reflect values relative to the control group within each experiment, and the representative plots display percentages out of TCRβ⁺ cells. Regarding the MMP9 inhibitor study (Fig. 6E–G), the treatment schedule was designed to pre-condition the tumor microenvironment and impair DC trafficking in advance of tdRT. While the inhibitor was initiated prior to radiation, its sustained activity overlaps with the window of tdRT-induced DC migration. This treatment strategy is consistent with prior studies showing that MMP9 inhibition can suppress immune activation by blocking dendritic cell trafficking to draining lymph nodes (He et al., *J Immunol* 2018; Yen et al., *Blood* 2008). We have clarified this rationale in the revised Methods section and updated the corresponding figure legends to indicate the normalized gating approach.

Minor Point 1: “Supp Fig 2C legend states ‘Tumor growth curves for control, targeted field, and large field tdRT-treated groups (4Gy, 8Gy, 12Gy tdRT).’ All figures are labeled tdRT but large field would encompass LNs—correct?”

We thank the reviewer for pointing this out. There was no large-field RT group in Supplemental Fig. 2C; all RT groups in this panel reflect tumor-directed (tdRT) fields only. The figure legend has been corrected to clarify this point.

Minor Point 2: “Supp Fig 2D—Is labeling of axes incorrect for the Texh and Tpex subsets?”

We appreciate the reviewer’s careful attention to detail. The axis labels in Supplemental Fig. 2D were incorrect and have now been corrected in the revised figure, now Supplemental F 1H.

Reviewer #2

The work presented demonstrates the role of the sentinel lymph node in priming of anti-tumor T cell responses in orthotopic preclinical models of HNSCC treated with focal radiotherapy. The pro-immunogenic effect of radiation is achieved with a single 4 Gy dose, which by itself has no effects on the tumor growth but can significantly increase the response to anti-PD1.

This work is a follow up to prior work by the authors and by others highlighting the importance of sparing draining nodes (and especially the sentinel node) when radiation therapy is used with the intent of increasing responses to immunotherapy. In this manuscript new mechanistic insights are provided about the role of the draining node in the pro-immunogenic effects of radiation therapy. The experiments are sound and elegant and leverage genetic and pharmacologic tools to present a detailed analysis of the role of dendritic cells that migrate between the tumor and the lymph node in the response to combinations of tumor-targeted radiation and anti-PD1. However, the translational significance and scientific novelty could be further enhanced in some areas, as detailed below.

Comment 1: “As recognized by the authors in Discussion, the use of a single 4 Gy dose is a limitation as this is not clinically relevant. The two clinical trials that inspired – in part – this work (refs. 7 and 8) use doses of 6 to 8 Gy repeated 2 to 5 times in combination with anti-PD1. Thus, it seems a missed opportunity not to investigate if using other fractionated regimens used in the clinic (6 Gy_x2 or 6Gy_x3 or 8Gy_x2 and 8Gy_x3) could increase cure rates with anti-PD1 above what is observed with 4Gy single dose, or rather, show potentially detrimental effects. One question that is extremely interesting and could be addressed using the R26-CreERT2 x Ai9 and local injection of tamoxifen is what is the optimal spacing between radiation doses (24 hours, 48 hours?) to enable the generation and migration of CCR7⁺ DC to the SLN.”

We thank the reviewer for this valuable suggestion, which helped refine the translational relevance of our study. In line with this recommendation, we focused on a monotherapeutic dose regimen (4 Gy × 1) that does not elicit a tumor response on its own (as observed with other monotherapeutic fractionation schemes tested Fig2D,

Supplemental Fig 2C), allowing us to isolate and characterize early immunologic effects of radiotherapy in the absence of confounding cytotoxicity. We then compared this to a 4 Gy × 2 regimen and observed a marked improvement in tumor control (Supplemental Fig. 2D). Although the fractionated regimens used in the clinic differ in dose and schedule, our model demonstrates that even modest alterations in dosing can significantly affect the host response. To address the possibility of detrimental effects, we refer the reviewer to our data in Figure 4E–F, where wide-field 4 Gy radiation (elective nodal irradiation) abrogates the therapeutic benefit of tdRT→αPD-1, underscoring the importance of lymphatic sparing.

To examine the impact of dose timing and spacing on mobilization of migratory dendritic cells, we used the R26-CreERT2 x Ai9 model with 1 or 2 fractions of 4 Gy and assessed CCR7+ dendritic cell migration to the SLN (Supplemental Fig 2E). Interestingly, although 4 Gy × 2 induced a complete tumor response, the magnitude of CCR7+ DC migration was comparable between the one- and two-fraction regimens. These data suggest that DC trafficking may saturate with a single dose, and that additional fractions likely engage non-redundant cytotoxic mechanisms that contribute to tumor rejection. This interpretation is consistent with recent observations reported in *Sci Transl Med* (PMID: 38222676), and we will discuss this further in the revised manuscript.

Comment 2: “Figure 1H and Supplemental Fig 1F show that the SLN immune migratome includes, besides DC, distinct populations of CD4+ T cells, CD8+ T cells, B cells, and NK cells. However, this important observation is not further developed. It seems important to know what the B cell clusters are – whether they could perform regulatory function – as previously described (PMID 37344597) or may serve as antigen-presenting cells. Similarly, the nature of the T cells that are labelled by tdTomato and move from the tumor to the SLN is of interest. What happens to these populations when the tumor is irradiated and/or treated with antiPD1?”

We thank the reviewer for highlighting this important observation and agree that deeper analysis of the migratory immune populations enhances the overall impact of our findings. In response, we performed CITE-seq analysis of tdTomato+ immune cells in the SLN without treatment, capturing the baseline migratory immune compartment, now shown in Main Fig. 1I, which enumerates and defines the composition of the migratory immune compartment. In response to the reviewer’s suggestion, we further analyzed the B cell compartment in Supplemental Fig. 1J, which revealed transcriptional signatures consistent with antigen-presenting and germinal center-like phenotypes but no evidence of regulatory B cell subsets. Finally, we examined how tdTomato+ CD8+ T cells respond to treatment in Supplemental Fig. 5H. Following tdRT→αPD-1, we observed increased frequencies of antigen-specific T cells in tumors (Figure 4C) in addition to enhanced expression of T cell programs of activation, migration and stem-like phenotypes in SLN (Supplemental Figure 5H).

Comment 3: “Figure 4B: is surgery immunosuppressive, at least temporarily? What happens if surgery is performed but the SLN not removed?”

We thank the reviewer for this important question. To address this directly, we performed experiments in which mice underwent surgical tumor resection while preserving the SLN. As shown in Main Fig 6A–D, this intervention did not impair the therapeutic response to tdRT→αPD-1, indicating that surgery itself is not immunosuppressive in this context. These findings are consistent with our previously published work, which similarly demonstrated that surgical resection alone does not diminish responsiveness to immune checkpoint blockade. Together, these results highlight that it is the removal of the SLN—not surgery per se—that impairs treatment efficacy.

Comment 4: “A minor issue: Supplementary Figure 5E mentioned on page 9 does not exist.”

We thank the reviewer for catching this oversight. The reference to Supplementary Figure 5E was made in error and has been removed from the revised manuscript. The revised manuscript now includes a new Supplemental Figure 5E, which is reference appropriately in the text.

Reviewer #1

Thank you to the authors for carefully addressing my critiques. With the exception of some needed clarification on a couple of my comments, authors have addressed all of my other critiques.

For Comment 3, I wanted clarification on whether injecting tamoxifen and letting it drain for 72 hours would result in labeling of tdTomato cells in the SLN. The experiment as presented shows that ovalbumin/STING are injected and 72 hours later tamoxifen is injected. How long after this 72 hr injection were the SLN analyzed with and without lymphatic channel ablation? Was this done 48-72 hours after tamoxifen as stated in methods line 698? Please include this information in the schema/ legend.

We thank the reviewer for the thoughtful follow-up and apologize for the confusion created by our original cartoon and legend. We agree that the timeline was unclear and appreciate the opportunity to clarify. In this experiment, local inflammation (OVA/STING) and tamoxifen were co-administered in the oral cavity. Seventy-two hours later, mice underwent either sham surgery or surgical ablation of the sentinel lymphatic channel, and SLNs were harvested 48 hours after surgery (i.e., 120 hours after tamoxifen and inflammation).

We have now revised the cartoon schematic in Figure 1C to accurately reflect this timeline and have updated both the figure legend and the Methods section to clarify the experimental sequence.

For Comment 6, I had requested gating controls for the H-2Kb-SIINFEKL antibody (25-D1.16) used in Figure 2H. I was not referring to the H2Kb-SIINFEKL tetramer specificity (as shown in Fig S2J). I think Lines 218-219 stating that the OT-1 T cell gating strategy in Fig S2J which shows tetramer specificity does not apply to the Figure 2H.

We thank the reviewer for this important clarification. We now understand that the original request referred specifically to gating controls for the H-2Kb-SIINFEKL antibody (25-D1.16) in Figure 2H, rather than the tetramer controls shown previously. To address this, we have expanded Supplemental Figure 2I to include the gating strategy for antigen-presenting dendritic cells, including H-2Kb-SIINFEKL⁺ populations shown in Figure 2H. We also moved the tetramer gating strategy originally in Figure S2J to Supplemental Figure 4C, where it now directly supports the OT1 T cell analyses in Figure 4C.

We have updated the figure legends and text accordingly.

Reviewer #2

The authors have satisfactorily addressed my prior questions and the manuscript is improved and overall provides important new information.